# Effective Decision Boundary Learning for Class Incremental Learning

Affiliations `emails`

## Abstract

Rehearsal approaches in class incremental learning (CIL) suffer from decision boundary overfitting to new classes, which is mainly caused by two factors: insufficiency of old classes data for knowledge distillation and imbalanced data learning between the learned and new classes because of the limited storage memory. In this work, we present a simple but effective approach to tackle these two factors. First, we employ **re**-sampling strategy and **M**ixup **K**nowledge **D**istillation (Re-MKD) to improve performances of KD, which would greatly alleviate the overfitting problem. Specifically, we combine mixup and re-sampling strategy to synthesize adequate data used in KD training that are more consistent with the latent distribution between the learned and new classes. Second, we propose a novel **I**ncremental **I**nfluence **B**alance (IIB) method for CIL to tackle the classification on imbalanced data by extending the influence balance method into the CIL setting, which re-weights samples by their influences to create a proper decision boundary. With these two improvements, we present the **E**ffective **D**ecision **B**oundary **L**earning algorithm (EDBL) which improves the performance of KD and deals with the imbalanced data learning simultaneously. Experiments show that the proposed EDBL achieves state-of-the-art performances on several CIL benchmarks.

## 1 Introduction

Applications of deep neural networks (DNNs) in a real world require the ability of the system to learn new classes incrementally [1, 2, 3]. However, DNNs typically suffer from drastic performance degradation on the previously learned tasks after learning new classes when the past data are unavailable, which is well documented as catastrophic forgetting [4, 5, 6]. Recently, many methods in CIL have been proposed to try to deal with this problem. Owing to their superior performances, **R**ehearsal-based approaches utilizing **KD** [7] (**RKD**) [8, 9, 10, 11, 12] have been widely applied in CIL. However, recent works revealed that the RKD methods suffer from decision boundary overfitting to new classes, which is referred to as the (task-recency) bias towards new classes [3, 8, 10, 13].

As shown in Fig. 1, basic RKD approaches typically have two training objectives: transferring the existing knowledge from the old model by KD and learning new classes via minimizing the KD loss [7] and the cross entropy (CE) classification loss. However, RKD approaches usually suffer from the decision boundary overfitting problem caused by insufficient KD and imbalanced data between the old and new classes.

**Problem of Insufficient KD.** In CIL setting, the stored exemplars of old classes are limited because of the small memory budget. It is well known that the high capacity of DNNs is sufficient to memorize the entire training data [14, 15], so RKD methods suffer from insufficient data for KD training. RKD methods commonly use the stored exemplars of old classes and data of new classes to compute the KD

---

[*]Use footnote for providing further information about author (webpage, alternative address)—*not* for acknowledging funding agencies.

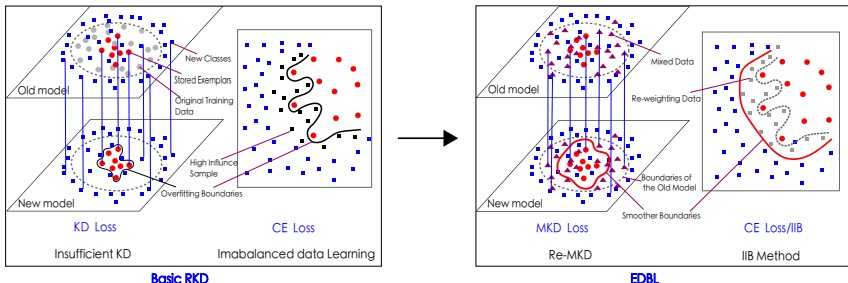

Figure 1: Comparisons between the basic RKD and the proposed methods (EDBL). Illustration of the two factors in RKD causing decision boundary overfitting to new classes.

loss to preserve the existing knowledge. However, data of new classes are out of distribution (OOD) with the original training dataset of the old model and Beyer et al. [16] demonstrated that OOD data work in KD to some extent while their performance suffers from great degradation, compared with using the original training data. The combination of the scantness of exemplars for the old classes and OOD between the learned classes and the new classes (data of old classes and new classes are typically OOD in the CIL setting) lead to the new model overfitting to new classes. This phenomenon is also demonstrated in our experiments (c.f. Sec. 5.5.2) and we refer to it as insufficient KD.

**Problem of Imbalanced Learning.** Due to the limited memory budget for storing exemplars of the old classes in RKD approaches, there exists a serious problem of imbalanced data for learning (few samples for the learned/old tasks are available while we have a large number of samples for the current/new task), which would lead to decision boundary overfitting to the dominant (new) classes [17, 18, 19].

In order to tackle the insufficient KD, we employ MKD and re-sampling strategy. We over-sample the exemplars of the old classes and mix the samples from the old classes and the new classes to synthesize mixed data for KD training. The interpolated data by an old class and a new class are more consistent with the distribution of the original training data of old classes than the data of new classes, which can improve KD in CIL and form a smoother decision boundary (c.f. Fig. 1, Re-MKD).

In order to deal with the imbalanced data learning problem, we attempt to attenuate the influence of samples that cause the overfitting. To this end, we extend the method of influence balance (IB) [19] to CIL setting and propose the incremental influence-balanced (IIB) method for CIL. Specifically, we first derive a metric that measures how much each sample influences the biased decision boundaries. Then, we decompose the metric into two parts which are referred to as the classification weighting factor and the KD weighting factor, respectively. Finally, we design the incremental influence-balanced (IIB) loss function for CIL, which adaptively assigns different weights to samples according to their influence on decision boundaries (c.f. Fig. 1, IIB Method).

Based on the previous proposal, we divide our method into two phases after re-sampling Mixup-based data augmentation. Phase 1 is MKD training, which improves the efficacy of KD to form a better decision boundary through generating sufficient and more related data with the old classes used in KD training. Phase 2 balances training by utilizing the derived IIB loss, attenuating the influence of the samples that cause overfitting of the decision boundary. Through the two training phases, we learn a decision boundary with better generalization capability.

In summary, the main contributions of this work are as follows: **(i)** We combine MKD with re-sampling strategy to relieve the insufficient KD problem in RKD methods through synthesizing data that are more related with the original training data of old classes than data of new classes. **(ii)** We extend the IB method into the CIL setting and propose a novel IIB loss function to tackle the imbalanced learning effectively in CIL. **(iii)** We propose the EDBL algorithm via combining Re-MKD and IIB by deriving the incremental influence factor for mixed data. Experiments demonstrate that the proposed EDBL method achieves state-of-the-art performances on several CIL benchmarks.

## 2 Related Work

### 2.1 Bias-Correction Approaches

CIL belongs to continual learning and the main challenge is catastrophic forgetting [4]. Previous approaches to tackle catastrophic forgetting can be divided into three categories [3]:regularization [20,

21], parameter isolation [22, 23] and replay approaches [24, 25]. RKD belongs to replay approaches and achieves the state-of-the-art performances. RKD methods [12, 9, 8, 10, 26] commonly train the new model by preserving the existing knowledge and learning classes incrementally via minimizing the KD loss and the CE loss. We introduce the problem formulation and summarize the popular strategies of RKD training in Appendix A.

RKD methods suffer from the task-recency bias and the ideas in previous works to alleviate this problem in CIL are very similar to the approaches in the long-tail learning. For example, iCaRL [12] employs representation learning strategy, and EEIL [9] utilizes classical image transformation e.g. random cropping, mirroring etc. to make data augmentation and fine-tune the newly trained model with a balanced dataset. BiC [8] trains a bias-correction classification layer with a balanced dataset, which is similar to the approaches of decouple learning and LUCIR [10] belongs to cost-sensitive learning. Different from the previous works, our method employs re-sampling and MKD strategy to relieve insufficient KD problem and propose IIB method to deal with the imbalance data learning. Both improvements are complementary to the previous bias-correction approaches.

## 2.2 Classes Imbalance

Classes Imbalance is a significant challenge for machine learning [27, 28]. In long-tail learning, the trained models usually bias towards the dominant classes. Previous efforts to tackle the overfitting problem can be roughly divided into three groups: (i) data re-balance including re-sampling [29, 30], data augmentation [31, 32], etc.; (ii) cost-sensitive learning including re-weighting strategy [33, 19], etc.; (iii) module improvement including representation learning [34, 35], decoupled training [36] and ensemble learning [37, 38], etc.. In this work, we combine mixup with re-sampling to make data augmentation and utilize the re-weighting strategy to relieve overfitting. But, we mainly use mixed data in KD training to relieve insufficient KD problem in CIL and we extend the re-weighting method in [19] (referred to as IB method in this work) into CIL and combine it with the re-sampling&mixup-based data augmentation to tackle imbalanced data learning.

## 2.3 Mixup and Knowledge Distillation

Mixup is first proposed in [39], which generates an interpolated sample $(\hat{x}, \hat{y})$ by $(x_i, y_i), (x_j, y_j)$ according to Eq. 1:

$$\hat{x} = \lambda x_i + (1 - \lambda)x_j, \hat{y} = \lambda y_i + (1 - \lambda)y_j, \tag{1}$$

where $x_i, x_j$ are images and $y_i, y_j$ are labels, respectively, $\lambda$ is randomly drawn by the Beta function. In [39], the CE loss ($L_{ce}$) for mixed data is computed as follows:

$$L_{ce}(\hat{x}, \hat{y}) = \lambda L_{ce}(\hat{x}, y_i) + (1 - \lambda)L_{ce}(\hat{x}, y_j). \tag{2}$$

After that, some other label mixing methods are proposed such as cutmix [40], manifold mixup [41], , Remix [31], etc.. Mixup-based data augmentation can greatly improve the generalization of DNNs in image classification learning while there are some works utilize label mixing methods in various scenarios such as the long-tail learning [31], continual learning [42, 43]. Recently, some works validate that label mixing methods improve the performance of KD e.g. [16, 44], which compute the KD loss with the mixed sample $(\hat{x}, \hat{y})$ as follows:

$$L_{kd}(\hat{x}, \hat{y}) = \sum_{i=1}^{m} -\sigma_i(\frac{T(\hat{x})}{t})log[\sigma_i(\frac{S(\hat{x})}{t})], \tag{3}$$

where $T, S$ are the teacher model and the student model, $t$ is the temperature, $m$ is the number of the learned classes of $T$ and $\sigma$ is the $softmax$. These works utilize Mixup to improve KD performances in model compressing setting where the training data are independent-identically-distributed (IID) with the original training dataset of the teacher model. However, they cannot verify the effectiveness of their methods in the CIL setting, where the training data are imbalanced and new classes are OOD with the old classes. In this work, we employ re-sampling in Mixup and validate effectiveness of mixup between old classes (IID dataset) and new classes (OOD dataset) for KD in the CIL setting.

## 3 Preliminaries

**Influence Function.** The influence function [45, 46] in robust statistics was proposed to find the influential instance of a sample to a model. Recent works have used influence function in DNNs,

e.g., [47]. Given a empirical risk $R(\Theta) = \frac{1}{N} \sum_{i=1}^{N} L(x, \Theta)$, of which the optimal parameter is $\Theta^* = argmin_\Theta R(\Theta)$, where $N$ is the number of training data, $\Theta = (\theta, W)$ are the parameters of the network and $\theta, W$ are the parameters of the feature extractor and the linear classifier (the last full connected layer, FC), respectively. According to the results in [47], the influence function of the point $(x, y)$ is given by:

$$\mathcal{I}(x, \Theta) = -H_L^{-1}(\nabla_\Theta L(x, \Theta)), \qquad (4)$$

where $\nabla_\Theta$ is the gradient operator and $H_L \triangleq \sum_{i=1}^{N} \nabla_\Theta^2 L(x_i, \Theta)$ is the Hessian, which is positive definite based on the assumption that $L$ is strictly convex in a local convex basin in the vicinity of $\Theta^*$.

**Influence Balance Method.** Park et al. [19] firstly apply influence function to a learning scheme, where they design the influence-balanced (IB) loss by utilizing the influence function during training in long-tailed classification. They treat the influential instance of a sample to a model of a point $(x, y)$ as the parameters of the model change when the distribution of the training data at the point $(x, y)$ is slightly modified and referred to it as the IB weighting factor. IB method uses $L_1$ norm of Eq. 4 to qualify the influence of the point $(x, y)$ and further ignores the inverse Hessian reasonably. The IB weighting factor is computed as follows:

$$\mathcal{IB}(x, \Theta) = \|\nabla_\Theta L(x, \Theta)\|_1 \qquad (5)$$

IB method focuses on the change in the FC layer of DNNs and the IB weighting factor can be simplified by:

$$\mathcal{IB}(x, \Theta) = \|f(x) - y\|_1 \|h\|_1 \qquad (6)$$

where $f$ is the network and $h$ is the hidden feature vector of $(x, y)$. Finally, the IB loss, which is used in balancing training to attenuate the influence of samples that cause an overfitted decision boundary is given by:

$$L_{IB}(x, y, \Theta) = \lambda_k \frac{L(x, y, \Theta)}{\mathcal{IB}(x, \Theta)} \qquad (7)$$

where in IB method, $L$ is CE loss function, $\lambda_k = \gamma n_k^{-1} / \sum_{i=1}^{K} n_i^{-1}$ is the class-wise re-weighting term, $k$ is the label of $(x, y)$, $n_i$ is the number of samples in the $k$-th class and $\gamma$ is the hyper-parameter. IB method has two training phases:**(i)** Normal classification training via minimizing the CE loss; **(ii)**Balancing training, where IB method fine-tunes the model via minimizing $L_{IB}$ by Eq. 7.

## 4 The Proposed Method

### 4.1 Re-sampling MKD

**Re-sampling Mixup.** Beyer et al. [16] demonstrates that KD training with OOD data suffers from great degradation and they validate empirically that the data, which are related or overlapped with the original training data (consistent with the latent distribution of original training data) can perform as good as the original training data in KD training. So the mixed data generated by mixup using samples from old classes and new classes, which are usually more related with old classes than the data of new classes (basic RKD methods directly use data of new classes in KD training) may improve KD performances in CIL.

Given that the data between the old and new classes are seriously imbalanced in RKD methods, we consider the ratio between the old classes from the past tasks and the new classes of the current task. We generate three kinds of mixed data in a batch: mixup among old classes, mixup between old classes and new classes and mixup among new classes. Owing to the limited data per old class in RKD methods, we mixup data more frequently for the first two types. Specially, we mixup $\frac{N}{2}$ times for the first two types and 1 time mixup for the last type of mixed data, where $N$ is the proportion of the data between per tail and head class, and we make sure the number of samples from old classes in a batch isn't less than a fixed number (32 in this work).

Finally, re-sampling-based Mixup can generate much data through label mixing between old classes and new classes. These generated data increases the diversity and the number of data used in KD, which improves the performance of KD in CIL. Our experiments demonstrate that re-sampling-based Mixup outperforms the original Mixup in [39] (referred to as Vanilla-Mixup in this work) significantly (c.f. Sec. 5.5.1).

**MKD Training.** After data augmentation by re-sampling-Mixup, we train the new model through minimizing two losses: the CE loss ($L_{ce}$) and the KD loss ($L_{kd}$) with the synthesized data in the

similar way as in the basic RKD method. We compute $L_{ce}$ and $L_{kd}$ according to Eq. 2 and 3, respectively and we replace the teacher and student model with the old model ($f_{\theta,W}^{t-1}$) and the new model ($f_{\theta,W}^{t}$) in Eq. 3, respectively. In this paper, we denote the strategy of Re-sampling Mixup and MKD training (Re-sampling MKD) as Re-MKD while we denote the strategy of vanilla Mixup and MKD training as vanilla-MKD.

## 4.2 IIB Method

### 4.2.1 IIB Weighting Factor

We extend the IB method into CIL and propose a novel IIB loss function to attenuate the influence of the samples to relieve the overfitting. The experience risk at task $t$ in RKD is $R(\Theta) = \frac{1}{N}\sum_{i=1}^{N} L(x, \Theta)$, where $L \triangleq L_{ce}(y_i, f_{\Theta}^{t}(x_i)) + L_{kd}(f_{\Theta}^{t-1}(x_i), f_{\Theta}^{t}(x_i))$ and $\Theta^* = argmin_{\Theta} R(\Theta)$ is the optimal parameter after initial training. We first utilize Eq. 4 to compute the influence of the sample $(x, y)$ for CIL. Then, like in IB method, we ignore the inverse of the Hessian, which is given by $H_L \triangleq \sum_{i=1}^{N} \nabla_{\Theta}^2 L(x_i, \Theta) = \sum_{i=1}^{N} \nabla_{\Theta}^2 L_{ce}(x_i, \Theta) + \nabla_{\Theta}^2 L_{kd}(x_i, \Theta)$ because it is commonly multiplied by all the training samples and just the relative influences of the training samples are needed and we use $L_1$ norm to qualify the influence to obtain the incremental influence-balanced (IIB) weighting factor. Therefore, according to Eq. 5 in the IB method, the IIB weighting factor of the point $(x, y)$ for CIL can be represented by:

$$\mathcal{IIB}(x, \Theta) = \|\nabla_{\Theta} L(x, \Theta)\|_1 = \|\nabla_{\Theta} L_{ce}(x, \Theta) + \nabla_{\Theta} L_{kd}(x, \Theta)\|_1 \tag{8}$$

At task $t$, let $h = [h_1, \ldots, h_L]^T$ be a hidden feature vector (an input to the FC layer), and $f(x, \Theta) = [f_1, \ldots, f_m, \ldots, f_{m+n}]^T$ be the output denoted by $f_k \triangleq \sigma(w_k^T h)$, where $\sigma$ is the $softmax$ function, $m, n$ are the number of learned classes and the new classes, respectively. The weight matrix of $g_W^t$ is denoted by $W = [w_1, \ldots, w_m, \ldots, w_{m+n}]^T \in R^{(m+n)\times f}$. Then, $\nabla_{\Theta} L_{ce}(x, \Theta)$ and $\nabla_{\Theta} L_{kd}(x, \Theta)$ is computed as below, respectively.

$$\frac{\partial L_{ce}(x, \Theta)}{\partial w_{kl}} = (f_k^t(x) - y_k)h_l, k \in [1, m + n] \tag{9}$$

$$\frac{\partial L_{kd}(x, \Theta)}{\partial w_{kl}} = \begin{cases} (f_k^t(x) - f_k^{t-1}(x))h_l, & k \in [1, m] \\ 0, & k \in [m + 1, m + n] \end{cases} \tag{10}$$

Finally, we apply Eq. 9 and 10 to Eq. 8. Thus, the IIB weighting factor can be computed as:

$$\mathcal{IIB}(x, \Theta) = (\|[2 * f^t(x) - y - f^{t-1}(x)]_1^m\|_1 + \|[f^t(x) - y]_{m+1}^{m+n}\|_1)\|h\|_1, \tag{11}$$

where $[V]_i^j$ denotes the slice $[v_i, \ldots, v_j](j \geq i)$ of the vector $V = [v_1, \ldots, v_N]$.

### 4.2.2 Decomposition of IIB Weighting Factor

**The Stability-Plasticity Dilemma.** The stability-plasticity dilemma [48] is a well-known constraint for artificial and biological neural systems. The basic idea is that a learning system requires plasticity for the integration of new knowledge, but also stability in order to prevent the forgetting of previous knowledge. Too much plasticity will result in previously encoded data being constantly forgotten, whereas too much stability will impede the efficient coding of this data at the level of the synapses.

In RKD, the KD loss is used to preserve the existing knowledge by encouraging the new model to mimic the output of the old model and the CE loss is used to learn to recognize new classes. Eq. 11 considers the influence of the CE loss and the KD loss on the decision boundary, but a sample may actually perform differently on the classification and KD training. For example, we use data of motorbike to train the new model to recognize motorbike incrementally while preserving the knowledge of knowing bike. Because of the similarity between motorbike and bike, samples of motorbike may improve the performance of KD and preserve the previous knowledge well. That is because the data which are related or overlapped with the original training data perform well in KD [16]. But in the classification training, the new model may bias to motorbike.

**Decomposing IIB Factor.** Considering the stability-plasticity dilemma, we decompose the IIB weighting factor into two factors: the classification weighting factor for learning new tasks and the KD weighting factor for preserving the existing knowledge, which measure how much each sample influences on forming new boundaries by the classification training and the previous decision boundaries of the old model preserving training (KD training), respectively.

**Definition 4.1.** In CIL, the classification weighting factor and the KD weighting factor of a sample $(x, y)$ are defined as the magnitude of the gradient vector of the CE loss and the KD loss on that point, respectively:

$$\mathcal{IB}_{ce}(x, \Theta) = \|\nabla_\Theta L_{ce}(x, \Theta)\|_1, \mathcal{IB}_{kd}(x, \Theta) = \|\nabla_\Theta L_{kd}(x, \Theta)\|_1 \qquad (12)$$

Then, we use a hyper-parameter $\alpha$ to make trade off on these two factors. Therefore, IIB weighting factor is computed via:

$$\mathcal{IIB}(x, \Theta) = \mathcal{IB}_{ce}(x, \Theta) + \alpha \mathcal{IB}_{kd}(x, \Theta) = (\|[f^t(x) - y]\|_1 + \alpha\|f^t(x) - f^{t-1}(x)\|_1)\|h\|_1 \qquad (13)$$

where $\mathcal{IIB}(x, \Theta) \leq \|\nabla_\Theta L_{ce}(x, \Theta)\|_1 + \|\nabla_\Theta L_{kd}(x, \Theta)\|_1$, and $\alpha \leq 1$.

### 4.2.3  IIB Loss

During the balancing training, we attempt to fine-tune the decision boundary to create a more generalized one, so we multiply the CE loss function by the inverse of IIB weighting factor, resulting in IIB loss as follows:

$$L_{IIB}(x, y, \Theta) = \lambda_k \frac{L_{ce}(x, y, \Theta)}{\mathcal{IIB}(x, \Theta)} \qquad (14)$$

where $\lambda_k$ is the class-wise re-weighting term (c.f. Sec. 3 and Eq. 7).

### 4.3  EDBL Algorithm

We propose the EDBL algorithm to combine Re-MKD and IIB method to improve knowledge transferring and deal with imbalanced data classification synthetically to generate more generalized decision boundaries in CIL. After MKD training, considering an mixed sample $(\hat{x}, \hat{y})$, we apply Eq. 1 to Eq. 13 to compute the IIB weighting factor as follows:

$$\mathcal{IIB}(\hat{x}, \Theta) = (\|[f^t(\hat{x}) - \lambda y_i - (1 - \lambda)y_j]\|_1 + \alpha\|f^t(\hat{x}) - f^{t-1}(\hat{x})\|_1)\|h\|_1 \qquad (15)$$

Then, we use this IIB weighting factor to compute $L_{IIB}$ according to Eq. 14.

Inspired by [16], which validates empirically that a large number of epochs improves the performance of KD, so we add the KD loss to the IIB loss. Then the overall loss $L_{overall}$ for balancing training is as follows:

$$L_{overall}(\hat{x}, \hat{y}, \Theta) = L_{IIB}(\hat{x}, \hat{y}, \Theta) + L_{kd}(\hat{x}, \hat{y}, \Theta) \qquad (16)$$

Overall, EDBL has two training phases: MKD training and balancing training. During the two training phases, $L_{kd}$ is continually used for distillation to encourage the decision boundary in the new model to mimic the one in the old model, and $L_{IIB}$ re-weights the interpolated samples by their influences on a decision boundary in the balancing training phase. The pseudo-code of EDBL is presented in Algorithm 1 in Appendix B.

## 5  Experiments

### 5.1  Datasets

We conduct experiments on CIFAR-10, CIFAR-100 [49] and Tiny-Imagenet [50] to validate our approach. We pad four pixels for images in CIFAR-10 and CIFAR-100 and then random crop them into $32 \times 32$ pixels. We use horizontal flip in all image pre-processings [12, 8].

### 5.2  Baselines and Protocols

We compare EDBL with some SOTA methods in CIL such as Mnemonics [51], PODNet-CNN [52] and SS-IL [26], etc.. We adopt two protocols to conduct experiments: (1) Base-half: we start from a model trained on half of all classes in the dataset, and the remaining half of classes come in 5 and 10 phases and stores 20 samples for each old class [10]. (2) Base-0: we follow the protocol in iCaRL [12] to conduct experiments, where all of classes come in 2, 5, or 10 phases and the memory budget is a fixed value. We report the results of two inference strategies: CNN output and the nearest-mean-of-exemplar strategy (NME) [12] to evaluate our methods (denoted as EDBL-CNN and EDBL-NME, respectively).

We follow the exemplar management of iCaRL to select exemplars for each old class. We use two common metrics: average accuracy and average incremental accuracy [12] to measure performances.

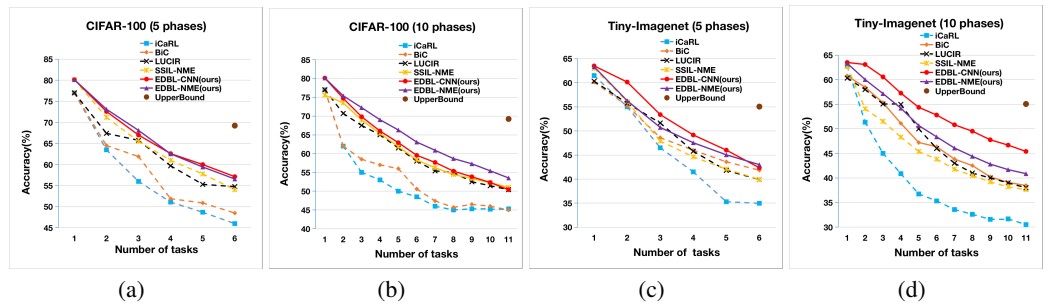

(a)       (b)       (c)       (d)

Figure 2: Comparison results of average accuracy on Base-half experiments with 5, 10 phases on CIFAR-100, Tiny-Imagenet.

## 5.3 Implementation Details and Hyper-parameters Tuning

The $\lambda$ in Eq. 1 is randomly sampled from the Beta function $B = (1,1)$ for all the experiments. To make comparison fairly, we use the same networks for our methods and baselines. We follow [12, 8, 10] and [43] to use ResNet32 (d=64) for experiments on CIFAR-10/100 and ResNet18 (d=512) for Tiny-Imagenet in this work. When training networks, we follow the standard practices for fine-tuning existing networks. We use stochastic gradient descent (SGD). As for the training hyper-parameters such as batch size, weight decay, momentum, learning rate, epochs etc., we set these parameters semi-heuristically and tune a few of parameters on CIFAR-100 and Tiny-Imagenet experiments with 5 incremental learning phases, then we use the same setting of these parameters for other experiments on CIFAR-10/100 and Tiny-Imagenet, respectively. We mainly tune $\lambda_k$ in Eq. 14 and $\alpha$ in Eq. 15 by grid searching. All the hyper-parameters are given in Appendix C. The code will be available at https://github.com/xxxx/CIL-EDBL.

## 5.4 Results

### 5.4.1 Results on Base-half

Table 1 presents the comparisons of our methods with the baselines on Base-half experiments. From Table 1, we can find that EDBL-NME outperforms all the compared methods on all the experiments significantly. EDBL-CNN also surpasses all the compared methods on all the experiments except the experiment on CIFAR-100 with 10 phases. The comparison results of average accuracy at each incremental learning phase are given in Fig. 2. Fig. 2 demonstrates that our methods (EDBL-CNN and EDBL-NME) surpass all the baselines at nearly each incremental learning phase.

### 5.4.2 Results on Base-0

We further conducted experiments on CIFAR-10/100 following Base-0 protocol to evaluate our method. The memory budgets on CIFAR-10 and CIFAR-100 are fixed to 200 and 2000 [12], respectively. CIFAR-10 is split into 2 and 5 phases while CIFAR-100 is split into 2, 5 and 10 phases. Our methods employ Mixup strategy which may help to relieve the overfitting problem while Remix [31] adapted Mixup to generate more effective interpolated data to tackle the long-tail learning. To compare our method with Remix, we directly employ Remix to train the new model to learn new classes

Table 1: Average incremental accuracy (%) on Base-half experiments. Models with an asterisk * denotes using the results in [51], The two models with † or ‡ are reported directly from [53] and [52], respectively. Because SS-IL [26] with NME performs much better than original SS-IL in this work, we alternatively select SS-IL with NME (SSIL-NME) as a baseline.

| Base-half | CIFAR-100 | | Tiny-imagenet | |
|---|---|---|---|---|
| Phases | 5 | 10 | 5 | 10 |
| LwF* [54] | 49.59 | 46.98 | / | / |
| iCaRL* [12] | 57.12 | 52.66 | / | / |
| BiC* [8] | 59.36 | 54.2 | / | / |
| LUCIR* [10] | 63.17 | 60.14 | / | / |
| PODNet-CNN‡ [52] | 64.83 | 63.19 | / | / |
| Mnemonics* [51] | 63.34 | 62.28 | / | / |
| TPCIL† [53] | 65.34 | 63.58 | / | / |
| iCaRL [12] | 59.67 | 56.13 | 48.98 | 39.27 |
| BiC [8] | 61.14 | 58.4 | 49.23 | 47.67 |
| LUCIR [10] | / | / | 49.31 | 47.56 |
| SSIL-NME [26] | 64.94 | 60.99 | 48.93 | 45.74 |
| EDBL-CNN(ours) | 66.57 | 62.06 | **52.43** | **53.80** |
| EDBL-NME(ours) | **66.65** | **64.73** | 50.99 | 49.97 |

Table 2: Average incremental accuracy (%) on Base-0 experiments.

| Base-0 | CIFAR-10 | | CIFAR-100 | | |
|---|---|---|---|---|---|
| Phases | 2 | 5 | 2 | 5 | 10 |
| iCaRL [12] | 89.7 | 77.13 | 68.35 | 67.24 | 63.98 |
| BiC [8] | 89.29 | 78.5 | 70.15 | 68.06 | 65.9 |
| PODNet-CNN [52] | / | 76.27 | / | 66.72 | 57.88 |
| RemiX-CNN [31] | 78.52 | 70.36 | 64.98 | 64.37 | 60.85 |
| RemiX-NME [31] | 82.07 | 77.41 | 67.19 | 67.18 | 64.46 |
| SSIL-NME [26] | 89.97 | 77.9 | 68.7 | 65.83 | 56.11 |
| EDBL-CNN(ours) | **91.6** | 77.42 | **72.28** | **72.2** | 66.53 |
| EDBL-NME(ours) | 89.82 | **78.51** | 70.71 | 69.33 | **67.99** |

incrementally with the optimal hyper-parameters given in [31]. We report the results of Remix-CNN and Remix-NME. From Table 2, we can find that EDBL-CNN and EDBL-NME outperform all

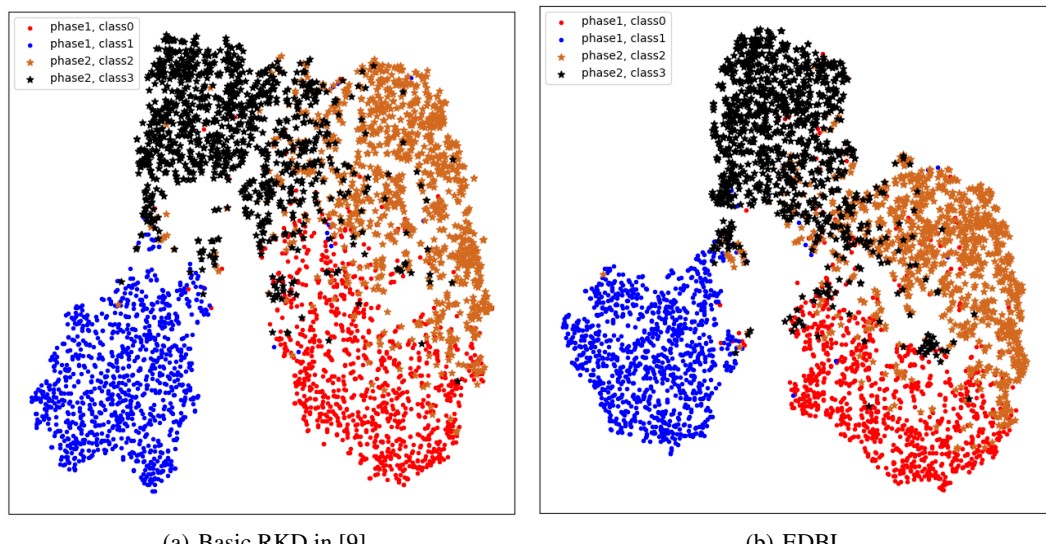

| (a) Basic RKD in [9] | (b) EDBL |

Figure 3: t-SNE visualization on a random subset (4 classes) from CIFAR-10 (2 incrmental phases).

Table 3: The effectiveness of Re-MKD&IIB of EDBL (Average Accuracy on each incremental phase).

| Dataset | CIFAR-10 | | | | | CIFAR-100 | | | | |
|---|---|---|---|---|---|---|---|---|---|---|
| phase | 1 | 2 | 3 | 4 | 5 | 1 | 2 | 3 | 4 | 5 |
| Baseline | 0.9894 | 0.775 | 0.5544 | 0.464 | 0.4427 | 0.848 | 0.7085 | 0.5606 | 0.4891 | 0.392 |
| + Vanilla-MKD | 0.9905 | 0.8095 | 0.587 | 0.4894 | 0.4189 | 0.8355 | 0.718 | 0.5658 | 0.4836 | 0.3625 |
| + IIB | 0.9894 | 0.8527 | 0.6678 | 0.558 | 0.5183 | 0.848 | 0.6914 | 0.5851 | 0.5068 | 0.4476 |
| + Re-MKD | 0.9905 | 0.8387 | 0.6446 | 0.6066 | 0.5926 | 0.8624 | 0.761 | 0.5744 | 0.5882 | 0.5629 |
| + IIB-KD | 0.989 | 0.819 | 0.7143 | 0.6782 | 0.6329 | 0.8624 | 0.7355 | 0.636 | 0.5788 | 0.5362 |
| EDBL-CNN(ours) | 0.992 | **0.8727** | **0.7245** | **0.6794** | **0.6618** | 0.85 | **0.767** | **0.7093** | **0.6573** | **0.6051** |

baselines on nearly all experiments except the experiment with 5 phases and 2 phases on CIFAR-10, respectively. Both of EDBL-CNN and EDBL-NME outperform Remix-CNN and Remix-NME on all the experiments by large margins about [1.1%, 7%]. We also give the results of average accuracy at each incremental learning phase and the results show that our methods surpass all the baselines at nearly each incremental learning phase, similarly (c.f. Appendix D).

## 5.5 Ablation Study

### 5.5.1 Analysis on Re-MKD and IIB

The base work in [9] is a vanilla basic RKD which trains the new model via minimizing the KD loss and the CE loss without any other strategies. We use it as the baseline in ablation study. The memory size in the ablation study is fixed to 200 and 2000 for CIFAR-10/100, respectively and the experiments are conducted in the same way as in the Base-0 experiments. Table 3 demonstrates the effectiveness of Re-MKD and IIB components in the proposed EDBL algorithm, where IIB-KD is the training strategy that uses the IIB loss and the KD loss to fine-tune the new model in the balancing training phase.

In summary, we can observe that: (1) The Re-MKD component can improve the performance of the baseline at each incremental batch by large margins, about [1%, 16%] and Re-MKD outperforms Vanilla-MKD (vanilla-Mixup-based MKD) significantly. This validates that the generated data by re-sampling-based Mixup are more related with old classes to improve the performance of KD. This also demonstrates that the Re-MKD would relieve the problem of insufficient KD; (2) The IIB component can improve the performance of the baseline at each incremental batch remarkably, and surpasses baseline by up to [5.5%, 7.5%] on CIFAR-10 and CIFAR-100 at the last phase. Besides, IIB-KD further improves the performance based on the improvement by IIB, which exceeds baseline by 19% and 13% on CIFAR-10 and CIFAR-100, respectively at the last phase. All of these reflect the

effectiveness of IIB loss for CIL; (3) EDBL achieves the best performances, which implies EDBL can learn more generalized decision boundaries effectively.

Further, we conduct visualization experiments with 2 incremental learning phases on a randomly selected subset with 4 classes from CIFAR-10 and the memory is fixed to 200. As shown in Fig. 3, the features extracted by the new model trained by EDBL has the less mutual intrusion between the learned classes (the first batch classes) and the new classes (the second batch classes), which reflects indirectly that the decision boundaries learned by EDBL are with more generalizability than the ones generated by the Basic RKD method.

### 5.5.2 Study on Memory Budget

We further study the sensitivity of EDBL on the size of memory budget by conducting experiments on CIFAR-100 with 5 phases following Base-0 protocol. The average accuracy on the last phase of the experiments are demonstrated in Table 4 and the table shows that EDBL-CNN obtains the best performances. CLDR [42]

Table 4: Sensitivity on memory (Average Accuracy on the last phase of the experiments), * denotes using the results in [42].

| Memory | 200 | 500 | 1000 | 2000 |
|---|---|---|---|---|
| GEM* [55] | 0.2829 | 0.3204 | 0.3546 | / |
| ER-Res.* [56] | 0.2869 | 0.3190 | 0.3406 | / |
| CLDR* [42] | 0.3434 | 0.3770 | 0.4020 | / |
| iCaRL [12] | 0.4189 | 0.4999 | 0.5133 | 0.5431 |
| SSIL-NME [26] | / | 0.4984 | 0.5217 | 0.5371 |
| EDBL-CNN(ours) | **0.4268** | **0.5265** | **0.5574** | **0.6051** |

utilizes manifold mixup to make a regularization to alleviate catastrophic forgetting while our method outperforms CLDR by large margins about [8%,15%].

### 5.5.3 Sensitivity of $\alpha$

In IIB method, we introduce the hyper-parameter $\alpha$ to trade off the classification weighting factor and the KD weighting factor to compute IIB loss. Here, we choose the strategy, which is RKD + IIB-KD in table 3 and $\alpha = \{1, 1e-3, 1e-5, 0\}$ to conduct experiments with 5 incremental phases on CIFAR-10/100 and the memory is fixed to 200, 2000, respectively.

Table 5: Sensitivity study on $\alpha$ (Average Accuracy on the last phases of the experiments).

| Dataset | CIFAR-10 | | | | CIFAR-100 | | | |
|---|---|---|---|---|---|---|---|---|
| $\alpha$ | 1 | 1e-3 | 1e-5 | 0 | 1 | 1e-3 | 1e-5 | 0 |
| CNN | 0.6034 | 0.596 | 0.6309 | 0.6176 | 0.4955 | 0.4977 | 0.5018 | 0.4715 |
| NME | 0.6485 | 0.6599 | 0.6577 | 0.6424 | 0.4836 | 0.4883 | 0.5016 | 0.5015 |

From Table 5, we can observe that IIB method is sensitive to the setting of $\alpha$ and setting a small value for $\alpha$ improves the performance significantly while $\alpha = 0$ is not the optimal value, where $\alpha = 0$ denotes that IIB degenerates into the IB method. This implies that stability-plasticity dilemma really exists in CIL and we should carefully treat this issue to achieve a better performance.

## 6 Conclusion

In this paper, we analyzed the causes of the decision boundary overfitting problem in CIL by two factors: insufficient KD and imbalanced data learning. We proposed the EDBL algorithm to address these problems. First, in order to deal with insufficient data for KD, we presented the re-sampling MKD strategy, which generates much data by Mixup and re-sampling to be used in KD training, which are more consistent with the latent distribution of old classes. Second, we extended IB method into CIL through deriving a novel IIB loss, which re-weights samples by their influence on decision boundary to alleviate the problem of learning with imbalanced data. On top of this, we propose EDBL algorithm to combine re-sampling MKD and IIB method which can improve the knowledge transferring and deal with the imbalanced data classification simultaneously. Experiments show that EDBL achieves state-of-the-art performances on several benchmarks and ablation study validates that both of Re-MKD and IIB are effective in CIL.

**Limitation and Broader Impacts.** One limitation is that more GPU memory and training time are needed because:(i) re-sampling Mixup generates much mixed data during training, (ii)EDBL has two training phases, so EDBL nearly requires twice as much training time as basic RKD method. Fortunately, both of re-sampling MKD or IIB are effective in CIL, so we can employ them in real applications flexibly. Moreover, EDBL significantly relieves catastrophic forgetting, which promotes the practical use of DNNs. The negative impacts may occur in some malicious or misuse scenarios. The appropriate proposes of employing EDBL (re-sampling MKD/IIB) are supposed to be ensured with attentions.

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

# Appendix

## A  Problem Formulation and RKD Methods

### A.1  Problem Formulation of CIL.

In the CIL setting, a dataset $\mathcal{D} = \{(x,y)|x \in \mathcal{X}, y \in \mathcal{Y}\}$ is split to $T$ subsets: $\mathcal{D} = \mathcal{D}^1 \cup \mathcal{D}^2 \cup \cdots \cup \mathcal{D}^t$, where $\mathcal{X}$ is a set of images with labels $\mathcal{Y}$ and the subsets have no overlapped classes, then a learning system is trained to learn each subset incrementally. At task $t$, we have a model $f_{\theta,W}^{t-1}$ which has incrementally learned the old classes $\tilde{\mathcal{C}}^{t-1} = \{\mathcal{C}^1, \mathcal{C}^2, \ldots, \mathcal{C}^{t-1}\}$, where $\theta, W$ denote the parameters of the feature extractor and the linear layer of the network, respectively and $\mathcal{C}^i$ denotes the classes in $i$ subset (task $i$). Now, given the new subsets $\mathcal{D}^t$ with the new classes $\mathcal{C}^t$, the goal is to train a new model $f_{\theta,W}^t$ that can perform classification on all the classes $\tilde{\mathcal{C}}^t = \tilde{\mathcal{C}}^{t-1} \cup \mathcal{C}^t$. In rehearsal-knowledge-distillation-based (RKD) methods, they store a small number of image exemplars of the old classes after the completion of each incremental learning task for experience replay at the future tasks. We denote $E^t$ as the selected exemplars of the current task (the new classes) to be stored after the completion of task $t$. We denote $\tilde{E}^t = \tilde{E}^{t-1} \cup E^{t-1}$, $\tilde{\mathcal{D}}^t = \mathcal{D}^t \cup \tilde{E}^t$, $\tilde{\mathcal{X}}^t = \{x|(x,y) \in \tilde{\mathcal{D}}^t\}$, $\tilde{\mathcal{Y}}^t = \{y|(x,y) \in \tilde{\mathcal{D}}^t\}$ as all the stored exemplars of the old classes, all the observable dataset, all the available images and labels at task $t$, respectively.

### A.2  Training Strategy of RKD Method.

Most previous works [12, 9, 8, 26] of RKD methods have the common process that uses all the available data to train the new model by minimizing two losses: the classification cross-entropy (CE) loss and the knowledge distilation (KD) loss. The CE loss is used to learn new classes and The KD loss is used to encourage the new network $f_{\theta,W}^t$ to mimic the output of the previous task model $f_{\theta,W}^{t-1}$. The CE loss ($L_{CE}$) and the KD loss ($L_{kd}$) are typically computed as follows:

$$L_{CE} = \sum_{(x,y) \in \tilde{\mathcal{D}}^t} \sum_{i=1}^{m+n} -\delta_i(x) log[\sigma_i(f_{\theta,W}^t(x))] \tag{17}$$

$$L_{kd} = \sum_{x \in \tilde{\mathcal{X}}^t} \sum_{i=1}^{m} -\sigma_i(f_{\theta,W}^{t-1}(x)) log[\sigma_i(f_{\theta,W}^t(x))]. \tag{18}$$

where $\delta_i(x)$ is the label indicator function, $m, n$ are the number of learned and new classes respectively and $\sigma$ is either the $softmax$ or $sigmoid$ function. So the new model $f_{\theta,W}^t$ are trained by the overall loss:

$$L = L_{kd} + \lambda L_{CE} \tag{19}$$

where $\lambda$ is the hyper parameter. Note that $f_{\theta,W}^t$ is continually updated at task $t$, whereas the network $f_{\theta,W}^{t-1}$ is frozen and will not be stored after the completion of task $t$.

However, the dataset of the new classes ($\mathcal{D}^t$) in the new task are out-of distribution (OOD) with the original training data ($\tilde{\mathcal{D}}^{t-1}$) of the old model $f_{\theta,W}^{t-1}$, so the performances of KD suffer from huge degradation. Moreover, RKD methods suffer from the task-recency bias [3]. After training the new model, to tackle the task-recency bias, various RKD methods have different subsequent processing. For example, iCaRL [12] takes the nearest-mean-of-exemplars (NME) classification strategy to make inference, BiC [8] trains a bias-correction layer with a balanced dataset and EEIL [9] further fine-tunes the whole model by using the balanced dataset of stored exemplars.

## B  EDBL Algorithm

The training process of our method (EDBL) is shown in **Algorithm 1**. At each incremental learning task, we first make data augmentation by re-sampling Mixup, then we train the new model with the mixed data in the same way as in the basic RKD method. At last, we fine-tunes the whole model by Eq. 16 in the balancing training phase.

---

**Algorithm 1** EDBL Algorithm for CIL

---

**Input:** Exemplars ($\tilde{E}^t, t > 1$) and data of new classes ($D^t$), $f^{t-1}$.
**Output:** Exemplars $\tilde{E}^{t+1} = \tilde{E}^t \cup E^t$, The New Model $f^t$
**Mixup with Re-sampling:**
Employ Mixup and Re-sample old classes to generate interpolated dataset $\hat{D}$.
**Phase 1: MKD Training**
**for** $i = 1$ **to** $T_1$ **do**
    sample a mini-batch $\hat{D}_m$ from $\hat{D}$
    $L_{ce} \leftarrow$ Eq. 2, $L_{kd} \leftarrow$ Eq. 3,
    $L(\Theta) = \frac{1}{m} \sum_{(\hat{x},\hat{y}) \in \hat{D}_m} L_{ce}(\hat{x}, \hat{y}, \Theta) + \gamma_1 L_{kd}(\hat{x}, \hat{y}, \Theta)$
    Update $\Theta^{t+1} = \Theta^t - \eta_1 \nabla L(\Theta)$
**end for**
**Phase 2: Balancing Training**
**for** $i = 1$ **to** $T_2$ **do**
    sample a mini-batch $\hat{D}_m$ from $\hat{D}$
    $\mathcal{IIB}(\hat{x}, \hat{y}, \Theta) \leftarrow$ Eq. 15
    $L_{overall}(\Theta) = \frac{1}{m} \sum_{(\hat{x},\hat{y}) \in \hat{D}_m} \lambda_k \frac{L_{ce}(\hat{x},\hat{y},\Theta)}{\mathcal{IIB}(\hat{x},\hat{y},\Theta)} + \gamma_2 L_{kd}(\hat{x}, \hat{y}, \Theta)$, $L_{ce}, L_{kd}$ is computed by Eq.2
    and 3
    Update $\Theta^{t+1} = \Theta^t - \eta_2 \nabla L_{overall}(\Theta)$
**end for**
**Exemplar Management:** Utilize the strategy in [12] to make exemplar management to select $E^t$ (maybe also remove some samples from $\tilde{E}^t$) to generate $\tilde{E}^{t+1}$ .

---

# C  Implement Detail

## C.1  Typical Training Hyper-parameters Selection

We draw $\lambda$ in Eq. 1 randomly from the Beta function B = (1, 1) for all the experiments. In re-ampling Mixup, we heuristically make sure the number of samples from old classes in a batch isn't less than 32. We semi-heuristically set the typical training hyper-parameters, e.g. epoch, learning rate, batch-size, etc. and tune a few of parameters on CIFAR-100 and Tiny-Imagenet experiments with 5 incremental learning phases, then we use the same setting of these parameters for other experiments on CIFAR-10/100 and Tiny-Imagenet, respectively. All experiments use the same batch size, weight decay, momentum: 128, 0.0002, 0.9, respectively. Other training details of the two stages are as below:

**MKD Training.** In Mixup-based Knowledge distillation (MKD) We train the new model with different hyper parameters on different datasets. For CIFAR-10/100, we train the network for 150 epochs at each task. The learning rate is set to 0.1, and reduced by a factor of 10 at 60, 100, 130 epochs. As for Tiny-Imagenet, the number of training epochs is 250 at each task. The learning rate is set to 0.1, and reduced by a factor of 10 at epochs 75, 125, 175 and 225.

**Balancing Training.** For CIFAR-10/100, the training epoch is 100, the learning rate is set to 0.01 and reduced by a factor of 10 at 30, 60, 80 epochs. For Tiny-Imagenet, the number of training epochs is 150 for each task. The learning rate is set to 0.01, and reduced by a factor of 10 at epochs 60, 100 and 130.

## C.2  Tuning on $\lambda_k$ and $\alpha$

We mainly make tuning on $\lambda_k$ in Eq. 14 and $\alpha$ in Eq. 16. $\lambda_k$ origins from the IB method and it is given by $\lambda_k = \gamma n_k^{-1} / \sum_{i=1}^{K} n_i^{-1}$, where $k$ is the label, $n_i$ is the number of samples in the $k$-th class and $\gamma$ is the hyper-parameter. We tune $\gamma$ and $\alpha$ by grid searching and adopt different values on different dataset and different experiments, which are given in Table 6.

Table 6

| Dataset | Experiments | $\gamma$ | $\alpha$ |
|---------|-------------|----------|----------|
| CIFAR-10 | Base-0-2 Phases | 10 | 1e-6 |
|          | Base-0-5 Phases | 100 | 5e-6 |
| CIFAR-100 | Base-0-2 Phases | 100 | 5e-6 |
|           | Base-0-5 Phases | 300 | 5e-6 |
|           | Base-0-10 Phases | 100 | 5e-6 |
|           | Base-half-5 Phases | 300 | 5e-6 |
|           | Base-half-10 Phases | 100 | 5e-6 |
| Tiny-Imagenet | Base-half-5 Phases | 10 | 1e-6 |
|               | Base-half-10 Phases | 10 | 5e-6 |

# D Comparison Results of Average Accuracy

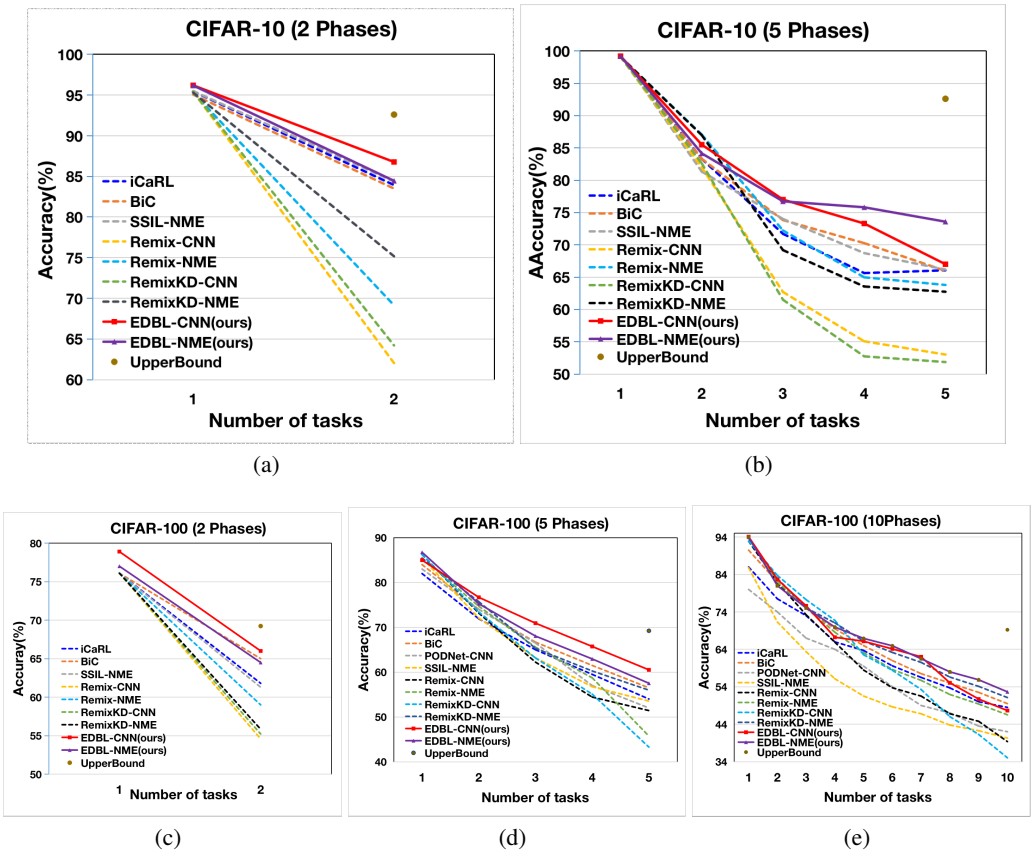

Figure 4: Comparison results of average accuracy of Base-0 experiments on CIFAR-10 with 2, 5 phases and CIFAR-100 with 2, 5, 10 pahses.

We conducted experiments on CIFAR-10/100 following Base-0 protocol to evaluate our method. The memory budgets on CIFAR-10 and CIFAR-100 are fixed to 200 and 2000, respectively. CIFAR-10 is split into 2 and 5 phases while CIFAR-100 is split into 2, 5 and 10 phases. Remix[31] adapted Mixup to generate more effective interpolated data to tackle the long-tail learning. Our method employs Mixup technique, so we use Remix as a compared method and directly employ Remix to train the new model to learn new classes incrementally with the optimal hyper-parameters given in [31]. In this supplement, to compare Remix with our method fairly, we further combine Remix with knowledge distillation (KD) to conduct experiments and report the results of CNN output and the nearest-mean-of-exemplar strategy (NME) (denoted as RemixKD-CNN and RemixKD-NME, respectively).

The comparison of the results are shown in Fig. 4. From Fig. 4, we can find that our methods (EDBL-CNN and EDBL-NME) outperform all the baselines nearly on every incremental task except EDBL-CNN loses to some baselines at the experiment on CIFAR-100 with 10 phases. Especially, our methods surpass Remix-CNN, Remix-NME and RemixKD-CNN, RemixKD-NME significantly by large margins about [1.1%, 22%] at the last incremental phase. We re-conducted the experiment of our methods on CIFAR-10 with 5 phases and we got better results, compared with the results given in Table 2. The incremental average accuracies of EDBL-CNN and EDBL-NME on CIFAR-10 with 5 phases are 80.4% and 81.894%, respectively, which surpass all the baselines significantly.

