# OpenReview forum: "Effective Decision Boundary Learning for Class Incremental Learning"
_NeurIPS.cc/2022/Conference — NeurIPS 2022 Submitted_

### Official Review · Reviewer_Fq7t · 2022-06-21

**Rating:** 4
**Confidence:** 3
**Soundness:** 2 fair
**Presentation:** 2 fair
**Contribution:** 3 good

**Summary:**

This paper first points out that the scantness of exemplars for the old classes and OOD between the previous classes and the new classes
cause insufficient knowledge distillation (KD). Then it proposes a two-phase method (EDBL) for class incremental learning. In the first phase, re-sampling strategy and Mixup Knowledge Distillation are used to generate mixed data and to form a better decision boundary.  The second phase uses the incremental influence-balanced (IIB) method to mitigate the imbalanced learning problem.  Experiments show that EDBL outperforms baselines by a large margin.

**Questions:**

Question 1:  The EDBL method generates more mixed data for the two types of mixup among old classes and mixup between old
classes and new classes than the type of mixup among new classes. But the rate ($N/2$) set for the first two types is just an empirical value. Please try to investigate the performance of EDBL at different rates and illustrate the results. Also, what does the "tail and head class" mean (Line 162)?

Question 2: Can the EDBL method help the KD method to alleviate the forgetting problem further? Please calculate and compare the average forgetting rate of the EDBL method and other baselines.

Question 3: The paper says the classification weighting factor is for learning new tasks. But it also calculates the classification weighting factor of the mixed data generated by the samples from old classes. Can the author explain the role of the classification weighting factor in this case?

Question 4: That is just a minor question: does the EDBL method also work well in the few-shot continual learning setting? In this setting, the model is over-fitting to the new data severely. And the IIB loss may have a significant impact in this setting.




**Limitations:**

I don't see any potential negative societal impact in this paper

**Strengths And Weaknesses:**

Originality: The idea of generating mixed data by mixup using samples from old classes and new classes has been used in class incremental learning [1].  So that is not a new contribution. The introduction of the Influence Balance method and the decomposition of the Incremental Influence Balance factor are novel. The novelty of this paper is somewhat reduced.

[1] Class-Incremental Learning via Dual Augmentation, NeurIPS2021.

Quality: This paper is technically sound and the claims are supported. However, the experimental setup is flawed. The authors should discuss and compare more batch continual learning methods which use the KD mechanism [2,3,4].

[2] Dark experience for general continual learning: a strong, simple baseline, NeurIPS2020

[3] Co2L: Contrastive Continual Learning, ICCV2021

The method outperforms many baselines in the setting of CIFAR100 and Tiny-ImageNet. But I don't know if the EDBL method also has the lowest average forgetting rate.

Clarity: This paper is well organized. I suggest the author give a simple introduction for task/domain incremental learning in Section 2 and a brief comparison of other approaches like the regularization-based approach.

Significance: The EDBL method outperforms many baselines in the setting of CIFAR100 and Tiny-ImageNet.  And the idea of calculating the incremental influence-balance weighting factor for each sample and then using the factor to adjust the loss is interesting.

---

> ### Author Response · Authors · 2022-08-02
> **Reply point-by-point to the comments**
>
> **Q1 (Compare more batch continual learning methods which use the KD mechanism (1,2,3): (1)DualAug(NeurIPS2021), (2)DER(NeurIPS2020), (3)Co2L(ICCV2021).)**
>
> Thanks to the reviewer's suggestion, we further conducted experiments according to the protocol in Co2L, applying the same model (ResNet 18) and the same data augmentation strategy. The results are shown in Tab.1 . From Tab.1, we can find that our proposed method significantly outperforms Co2L and DER by large margins on CIFAR-10 and Tiny-Imagenet, where EDBL surpasses the best baseline by about [3\%, 8\%] in Class-IL and about [1\%, 14\%] in Task-IL on CIFAR-10 and Tiny-Imagenet. We also compare our method with DualAug. From Tab.2, our method outperforms DualAug by large margins about [1\%, 7\%]. Please refer to Tabs.1 and 2 in the official comment of "Results of extra experiments and references for rebuttal " or the reply to the comments of reviewer 1CA1, "Reply point-by-point to the comments" for more detail.
>
>
> **Q2 (Study on re-sampling rate)**
>
> We further conducted a study on the re-sampling rate. We divide $N/2$ by the rates in the Tab.5 to conduct experiments on CIFAR-10 and CIFAR-100 with 5 phases in the Base-0 protocol. We report the results of EDBL-NME and also give the results of baseline-NME in Tab.5 . The result in Tab. 5 suggests that very large or low re-sampling rates are not beneficial. The "tail and head class" in (Line 162) denote the old and new classes because the stored exemplars of the old classes are limited while there are large-scale data for the new classes, so the training data are long-tailed.
>
> **Table 5. Results of study on the re-sampling rate.**
>
>     Dataset                  CIFAR-10                           CIFAR-100
>     phase          1     2      3     4       5        1      2      3      4      5
>     Baseline-NME 99.3  85.15  64.27 58.39	61.24     84.8  69.52  61.42  55.52  50.02
>           4      	               /                  84.8	69.77  63.4   58.2   53.91
>           2      99.5  79.72  71.68	69.48	66.9      84.8	68.27  63.32  58.75  54.47
>           1      99.5  86.87  75.33	74.14	71.27     84.8	72.17  65.05  58.99  55.27
>          0.5     99.5  87.32  76.48	70.54	68.64     84.8	72.25  64.95  58.34  54.14
>         0.25   	               /                  84.8	72.9   64.1   57.55  53.4
>
>  **Q3 (The average forgetting rate of the EDBL method and other baselines.)**
>
> We calculate the average forgetting rate~(FGT) of EDBL and other baselines, e.g. SSIL, iCaRL, BiC according to the FGT formulation given in {Overcoming Catastrophic Forgetting with Unlabeled Data in the Wild} (ICCV2019). The comparisons of FGT and average accuracy between them are shown in Tab. 6. As shown in Tab.6, the EDBL method especially EDBL-CNN has much lower average forgetting rates.  The FGT of EDBL-CNN are 6.45\%, 5.14\%, compared with iCaRL ( 20.98\%,11.15\%), BiC (19.58\%, 10.77\%), and SSIL-NME (8.55\%, 8.28\%)  on CIFAR-10/100, respectively.  EDBL also achieves the best average accuracy after completing the last learning phase.
>
> **Table 6. Average accuracy and Average forgetting rate (\%) after completing the last learning phase.**
>
>     Base-0           CIFAR-10  CIFAR-10    CIFAR-100  CIFAR-100
>       /                ACC       FGT 	   ACC	      FGT
>     iCaRL             66.06     20.98         54.2       11.15
>     BiC               65.97     19.58         56.5       10.77
>     SSIL-NME          66.17      8.55	  53.54	     8.28
>     EDBL-CNN(ours)    66.18      6.45	  60.51	     5.14
>     EDBL-NME(ours)    68.77	    12.90	  57.52	     12.92
>
>  **Q4 (The paper says the classification weighting factor is for learning new tasks. But it also calculates the classification weighting factor of the mixed data generated by the samples from old classes. Can the author explain the role of the classification weighting factor in this case?)**
>
> When learning new tasks, EDBL trains the new model by minimizing the classification cross-entropy loss in all the classes including the old and new classes. Because the training data are class imbalanced, EDBL computed the classification weighting factor to re-weight all the high-influenced samples to tackle the long-tail classification learning.
>
>   **Q5 (That is just a minor question: does the EDBL method also work well in the few-shot continual learning setting? In this setting, the model is over-fitting to the new data severely. And the IIB loss may have a significant impact in this setting.)**
>
> Thank you for the suggestion. This is an interesting issue,  we plan to apply EDBL to the few-shot continual learning scenario to study the effect of EDBL on the few-shot learning and report the results in the new version of the paper.

---

### Official Review · Reviewer_PDXX · 2022-07-06

**Rating:** 5
**Confidence:** 3
**Soundness:** 3 good
**Presentation:** 3 good
**Contribution:** 2 fair

**Summary:**

The authors address the problem of imbalance between old and new
classes in class-incremental learning due to the limited memory for
old classes.  For re-sampling, they added 3 types of Mixup instances:
among old classes, between old and new classes, and among new classes,
with proportionally more for the first 2 types to yield balanced
classes.  The loss function has cross entropy and knowledge
distillation.  They adapted influence balanced (IB) method to include
both the classification and knowledge distillation terms from the loss
function, which they call IIB.  They further use hyperparameter alpha
to weight the IIB factor for knowledge distillation (to decompose
knowledge distillation and cross entropy).  The loss function L_IIB is
lambda * L_ce / IIB, where lambda is class-wise reweighting.

The overall 2-phase EDBL algorithm combines mixing and IIB.  The first
phase has knowledge distillation.  The second phase has L_IIB, which
includes classification and knowledge distillation.

With 3 datasets, empirical results indicate the the proposed EDBL
generally outperforms a number of existing techniques.  An ablation
study also indicates the contribution of each of the proposed
components.


**Questions:**

According to Algorithm 1 in the Appendix, the main difference between
Phase 1 (MKD) and Phase 2 (Balancing training) is whether IIB is
incorporated or not.  Why not having only Phase 2?  A comparison of
the last 2 rows in Table 3 (if I understand correctly) seems to
indicate the benefit of adding Phase 1 (but seems to be not discussed
in Section 5.5.1).  Further insights would be important.

Eq 11: the first term seems to be missing 2* for f^t(x) and h. If so,
Eq 13 and Eq 15 seem to have similar issues.

Lines 223, 234, and 236: balancing training ->  balanced training?

Phases are used in both EDBL and CIL.  Using different terms could be
easier to separate the two concepts; e.g. stages in EDBL and phases in
CIL.


**Limitations:**

Limitations and negative societal impact were discussed.



**Strengths And Weaknesses:**

Strengths

The combination of Resampling with Mixup and IIB seem interesting.

Empirical results indicate the the proposed EDBL generally outperforms
a number of existing techniques.

Weaknesses

While the paper is generally well written, the motivation and
description of the overall 2-phase EDBL algorithm are not clear.  I
had to consult Algorithm 1 in the Appendix.  It could be improved by
describing and motivating the 2 phases first, and then the respective
loss functions that are involved.

Phase 2 can be considered as a superset of Phase 1--the motivation for
the need of Phase 1 could be further explained. (See questions below)

---

> ### Author Response · Authors · 2022-08-02
> **Reply point-by-point to the comments**
>
> **Q1 (Why adding Phase 1)**
>
> EDBL has two training stages. The first stage trains a new model by Re-MKD and fine-tunes it with the balanced training. Because the data of the added classes are OOD, the KD training in the first stage is not a typical long tail KD training~(Long tail KD training refers to distillation with long tail data, LT-KD). Thus, we apply the typical RKD method to train a new model and use Re-MKD to improve knowledge transferring. After we obtain a new model, the second training stage becomes a typical long-tail KD training, and we attempt to fine-tune it by tackling LT-KD. Thus, we compute the IIB factor, the KD weighting factor, to re-weight the high-influenced samples in the second training stage. Thanks to the reviewer's suggestion we further conducted experiments with only one training stage using IIB-KD. The results are shown in Tab. 4. From Tab. 4, we can find that directly using IIB-KD to train a new model performs worse than EDBL by a large margin.
>
> **Table 4. Results of Re-MKD + CBF on CIFAR-100 with 5 phases in Base-0 protocol~(Average Accuracy on each incremental phase, %).**
>
> 	Dataset                 CIFAR-100
>     phase                       1     2       3     4      5
>     BiC                       84.8  74.02  66.7   61.5   56.5
>     BiC+Re-MKD                84.8  71.73  59.36  57.59  53.51
>     EEIL                      83.5  76.5   64.2   59.1   52.8
>     EEIL+Re-MKD               84.8  71.85  64.78  58.14  52.84
>     IIB-KD(One-stage)         83.5  69.47  60.3   53.15  48.7
>     IIB-KD(One-stage)+Re-MKD  84.8  76.7   70.93  65.73  60.51
>
> **Q2 (Eq 11: the first term seems to be missing 2 x for $f^t(x)$ and h. If so, Eq 13 and Eq 15 seem to have similar issues.)**
>
>  We first add Eq.9 to Eq.10, then we apply the added result to Eq.8 then we get Eq.11, so Eqs.11, 13, and 15 are correct. Please refer to Eq.(1) in "response-5335.pdf" in the supplementary material for more detail.
>
> **Q3 (Lines 223, 234, and 236: balancing training -> balanced training? Phases are used in both EDBL and CIL. Using different terms could be easier to separate the two concepts; e.g. stages in EDBL and phases in CIL.)**
>
> Thank you for the comment. We will fix these minors and will apply all the suggestions to improve the readability of the new version of the paper.

---

> > ### Comment · Reviewer_PDXX · 2022-08-08
> > **Comments on author response**
> >
> > Q1:  I suggest you to discuss the motivation in Sec. 4.3
> >
> > Q2:  adding Eq 9 and 10 for k = 1 to m:  f^t(x) appears in both Eq 9 and 10, how did you reduce the sum to f^t(x), not 2*f^t(x)?

---

> > > ### Author Response · Authors · 2022-08-09
> > > **Thanks for Reviewer PDXX's feedback**
> > >
> > > - Q1. In Sec. 4.3, we attempt to use mixed data in IIB method. In Sec. 4.2, the IIB method is based on the raw images. In Sec. 4.3, we derive the formulation of IIB on the condition of mixed data by applying the formulation of Mixup (Eq.1) to IIB formulation (Eq.13) to get Eq. 15.
> > >
> > > - Q2. Thanks for the reminder, I double-checked the process of adding Eq.9 to Eq. 10, the first term of Eq. 11 should be $2 * f^{t}(x)$.  We have submited the corrected version, thanks again. But h in Eq. 11 is the common factor, which is taken out separately, so that is correct. As for Eq. 13,   the $IB_{ce}$ and $IB_{kd}$ are computed separately, they do not involve adding Eq.9 to Eq. 10， so f^{t}(x) must not be multiplied by 2. Eq.15 is based on Eq. 13, so $f^{t}$(x) must not be multiplied by 2 too. Specifically，we apply Eq.1 to Eq.13 to get Eq. 15, where $\lambda + 1 - \lambda = 1$ and there is no the process of adding Eq.9 to Eq. 10, so there is no 2 to be multiplied to  $f^{t}(x)$. Please note that our method uses Eq. 13 and Eq.15 but not Eq.11, so our implementation and results are correct.

---

> > > ### Author Response · Authors · 2022-08-09
> > > **The derivation of Eq. 13 and Eq. 15**
> > >
> > > Thanks for the comments again. Here, we give the process of deriving Eq. 13, that is $IIB = IB_{ce} + \alpha IB_{kd}$.
> > >
> > > $IB_{ce} = ||\triangledown L_{ce}||_{1}$
> > >
> > > $=||(\partial L_{ce} / w)||_{1}$
> > >
> > > $=||h(f^{t} - y)||_{1}$
> > >
> > > $IB_{kd} = ||\triangledown L_{kd}||_{1}$
> > >
> > > $=||(\partial L_{kd} / w)||_{1}$
> > >
> > > $=||h(f^{t} - f^{t-1})||_{1}$
> > >
> > > so, $IIB =(||(f^{t} - y)||_1 + \alpha * ||(f^{t} - f^{t-1})||_1)*||h||_1$
> > >
> > > so Eq. 13 is correct, Eq. 15 is based on Eq. 13, so it is correct too. Our method is based on Eq. 13 and Eq. 15, so our method is correct.

---

> > > ### Author Response · Authors · 2022-08-09
> > > **Why not use Eq. 11**
> > >
> > > - Eq.11 computes a unified weighting factor without considering the Stability-Plasticity Dilemma to balance the classification weighting factor and the KD weighting factor of a sample. As described in Sec. 4.2.2,  a sample really performs differently on the classification and KD training. Actually, using the unified weighting factor by Eq.11 performs badly in our experiments, so we decompose the IB factor into $IB_{ce}$  and  $IB_{kd}$ representing stability and plasticity factors, respectively. The explanation is important and clear to tackle with the problem of stability-plasticity dilemma.  It is noted that our approach trade-offs on Stability-Plasticity by computing a balanced reweighting factor, which is different from the previous approaches that directly balance between $L_{ce}$ and $L_{kd}$.

---

### Official Review · Reviewer_JLqd · 2022-07-08

**Rating:** 5
**Confidence:** 5
**Soundness:** 3 good
**Presentation:** 2 fair
**Contribution:** 3 good

**Summary:**

This paper addresses the problem of class incremental learning with an emphasis on adjusting the decision boundary. The authors claim that the deficiency of old data and the imbalanced training mainly cause the ineffective decision boundary. The paper proposes Re-MKD, which combines re-sampling strategy and Mixup, and Incremental Influence Balanced (IIB) loss to tackle the aforementioned problem. The experimental results on various benchmarks show the effectiveness of the proposed framework.

**Questions:**

- As mentioned in the weakness section, I wonder how the overall incremental learning performance would be affected when we exclude the mixed data of the newly added classes when computing $L_{kd}$.
- Why is the IIB factor only multiplied to $L_{ce}$? Since $L_{kd}$ also can affect the overall decision boundary and IIB factor also considers $L_{kd}$ as in eq (10), IIB weighting factor may be helpful to $L_{kd}$ in eq (16), too. If possible, it would be nice to see how the final performance will be affected if IIB factor is also considered to $L_{kd}$. Or, if I missed something, please let me know.
- Conventionally, many CIL methods [9, 10, 52] utilize additional “class balanced fine-tuning (CBF).” It seems that the second phase of EDBL algorithm, which is balanced training, can be a good replacement for CBF, since CBF only utilizes small number of examples unlike the proposed framework. Thus, it would be nice if the ablation study of balanced training is added to the paper. Or, is “Re-MKD” row in Table 3 indicates the Re-MKD + CBF?
- Figure 3 can be improved if it contains more classes than two classes. Even though it is a minor issue for me, it seems demonstrating only a single case may weaken the claim of the paper.

**Limitations:**

The paper sufficiently deals with the limitation of the paper and broader impacts.

**Strengths And Weaknesses:**

- Strengths
    - Overall, the paper is well-written and clear. The authors’ claim that imbalanced training is critical issue for the ineffective decision boundary is well supported throughout the paper. Both proposed methods (Re-MKD and IIB loss) may not be super-novel, but they are reasonable solution for the problem. In particular, the motivation of the proposed solutions are clear. Re-MKD makes good use of the rate rebalancing of the old data and can populate many diverse examples from the old data. IIB also seems suitable for the class incremental learning since class incremental learning can be viewed as long-tail problem, where the old data is extremely scarce.

- Weaknesses
    - I have an issue with the claim made in the paper, which leave some room for improvement.
        - The authors repeatedly emphasize that data of newly added classes, which are OOD data from the perspective of the previously learned classes, harm the knowledge distillation process. However, the proposed framework still uses the mixed data of the newly-added data for knowledge distillation. It would be nice if the paper includes the ablation study about the 3 types of mixing classes mentioned in Line 158-159. In other words, I wonder how the overall incremental learning performance will be affected when we exclude the mixed data of the newly added classes from computing $L_{kd}$.
    - There are some serious formatting issues throughout the paper.
        - Font size of the captions and equations is significantly smaller than the original submission format. This should be revised.
    - Minor notation error
        - The notation $L$ is used multiple times in different contexts. In general, $L$ denotes the loss, but in Line 188, $L$ denotes the feature dimension. Moreover, $f$ in Line 191 also needs to be corrected since $f$ does not indicate the dimension of the feature vector.

---

> ### Author Response · Authors · 2022-08-02
> **Reponse point-by-point to the comments**
>
> **Q1 (It would be nice if the paper includes the ablation study about the 3 types of mixing classes mentioned in Line 158-159. In other words, I wonder how the overall incremental learning performance will be affected when we exclude the mixed data of the newly added classes from computing $L_{kd}$.)**
>
> On the one hand, the stored exemplars of the old classes are consistent (independent-identically-distributed, iid) with the original training data of the old model, so they are used in KD training to preserve the old knowledge. On the other hand, Beyer et al. [16] demonstrate using OOD data has some positive effects on KD to some extent and most KD-based methods in CIL use the data of the added classes to make transferring the old knowledge. Based on the reviewer's comment, we further conducted experiments excluding the
> mixed data of the newly added classes from computing $L_{kd}$. As shown in Tab.3, when excluding the mixed data of the newly added classes in KD training, both the performances of baseline and EDBL-NME degrade drastically. From our experiments, using the data of the added classes improves the performance of KD in CIL. However, Beyer et al. [16]  also validate empirically that the effect of using the OOD data is very limited, compared with using the original training data for KD. So based on this observation, we focus on improving the effectiveness of using OOD data in KD training by re-sampling and Mixup.
>
> **Table 3. Results of experiments on excluding the mixed data of the newly added classes in KD training.**
>
> 	Dataset                                CIFAR-100
>     phase                     1     2       3     4       5
>     Baseline-NME             84.8  73.3   65.67  59.09  54.47
>     Baseline-NME +exclude    84.8  69.87  62.90  56.51  52.19
>     EDBL-NME                 85.01 75.25  68.05  62.95  57.52
>     EDBL-NME +exclude        84.8  63.95  58.38  52.62  48.55
>
> **Q2 ( multiply IIB factor to $L_{kd}$.)**
>
> Thank you for the comment. In our experiments, when multiplying IIB factor to $L_{kd}$, IIB-KD methods achieve average accuracy of 52.53\% after completing the last training phase on CIFAR-100 with the Base0 and 5 phases protocol, compared to 53.62\% without multiplying IIB factor to $L_{kd}$. Based on these results, we only multiply IIB factor to $L_{ce}$.
>
> **Q3 ( Study on Re-MKD + CBF.)**
>
> Both EEIL and BiC apply the CBF technique and they have two training stages, where EEIL utilizes classical image transformation to make data augmentation and then fine-tune the newly trained model with a balanced dataset while BiC trains a bias-correction classification layer with a balanced dataset. We conduct experiments with Re-MKD + EEIL and Re-MKD + BiC on CIFAR-100 with Base0 and 5 phases protocol to study the effect of Re-MKD on the CBF technique. The results shown in Tab.4 suggest that Re-MKD has different effects on different CBF techniques. Re-MKD improves the performance of IIB-KD but it does not improve the effect of EEIL and BiC.  Please refer to Tab. 4 in the "response-5335.pdf" provided in the Supplementary material for more information.
>
> **Table 4. Results of Re-MKD + CBF on CIFAR-100 with 5 phases in Base-0 protocol~(Average Accuracy on each incremental phase, %).**
>
> 	Dataset                 CIFAR-100
>     phase                       1     2       3     4      5
>     BiC                       84.8  74.02  66.7   61.5   56.5
>     BiC+Re-MKD                84.8  71.73  59.36  57.59  53.51
>     EEIL                      83.5  76.5   64.2   59.1   52.8
>     EEIL+Re-MKD               84.8  71.85  64.78  58.14  52.84
>     IIB-KD(One-stage)         83.5  69.47  60.3   53.15  48.7
>     IIB-KD(One-stage)+Re-MKD  84.8  76.7   70.93  65.73  60.51
>
> **Q4 (Formatting issues throughout the paper and Minor notation error )**
>
> We appreciate the reviewer's suggestion. We will provide visualization results with more classes and more incremental learning phases, and fix the minors to improve the presentation accordingly.

---

> > ### Comment · Reviewer_JLqd · 2022-08-06
> > **Thanks for the responses.**
> >
> > Thanks for the responses. However, some of my major concerns are still not fully addressed.
> >
> > - Q2) IIB Factor
> >     - Although I appreciate the empirical results, I wonder about the authors’ **interpretation** of the results, what might cause such degradation when $L_{kd}$ is weighted on the proposed IIB weighting.
> > \
> > &nbsp;
> >
> > - Q3) I guess the authors misunderstood what I asked. What I wonder is whether the proposed **balancing training (phase 2) can replace the CBF** process, not the **compatibility of Re-MKD (phase 1) and CBF (or Bias Correction)**. To be specific, in the first phase, train EEIL (or other baselines like LUCIR or POD) as the original one, and then compute IIB Loss in the second phase with the mixed samples. Nevertheless, I also appreciate the newly conducted experiment which provides another aspect of the proposed Re-MKD (phase 1).

---

> > > ### Author Response · Authors · 2022-08-07
> > > **Thanks for Reviewer  JLqd's feedback**
> > >
> > > Thanks for the comments. We reply point-by-point to the concerns as below.
> > >
> > > **Q2) Explanation on why $L_{kd}$ is not weighted on IIB factor**
> > > - In KD-based methods for CIL, $L_{kd}$ is used to transfer old knowledge while $L_{ce}$ is mainly used to learn the added classes. The Stability-Plasticity Dilemma suggests that we should balance between  $L_{kd}$ and $L_{ce}$ in the overall loss function. As the stored exemplars of the old classes are limited while there is a large scale of data for the new classes, the new model tends to bias towards the new classes, so the weight of $L_{kd}$ in the total loss function should be set relatively large. According to Eq. 13, that is $IIB=IB_{ce} + \alpha * IB_{kd}$, if $L_{kd}$ is weighted on the IIB factor, then the larger $\alpha$ is, the smaller the weight of $L_{kd}$ in the total loss function, so $\alpha$ should be set to a relatively small value. This argument is also verified by our experiments. Actually, $\alpha$ is set to a very small value, which is no more than $10^{-6}$  in all of our experiments (Please refer to Appendix C.2). Based on this observation, we can find that the main component of $IIB$ is $IB_{ce}$ and the key role in the IIB factor is the classification weighting factor ($IB_{ce}$), so we argue that it is reasonable that $L_{ce}$ is divided by IIB while $L_{kd}$ is not weighted on IIB. Please note that  $IB_{kd}$ still works in $L_{ce} / IIB$, as $\alpha=0$ is not the optimal value in the experiments (please refer to Sec. 5.5.3).
> > >
> > > **Q3) Experiments on balancing training (phase 2) replacing the CBF**
> > >
> > >  Thanks for the reminder. We misunderstood what you asked and we conduct the experiments on balancing training (phase 2) replacing the CBF.  SS-IL proposes task-wise KD to tackle LT-KD while iCaRL applies a representation learning strategy. We think that the IIB method (phase 2) can improve task-wise KD and representation learning, so we conduct experiments on iCaRL + IIB and SS-IL +IIB to verify whether IIB can replace CBF.  At the first training stage, we apply iCaRL or SS-IL to train the new model, then we fine-tune the new model by the IIB method (the balanced training) at the second training stage. The buffer size is fixed to 2000 as in iCaRL[12]. The results are  shown in Tab. 7.
> > >
> > > **Table 7. Results of iCaRL + IIB and SS-IL +IIB on CIFAR-100 with 5 phases in Base-0 protocol~(Average Accuracy on each incremental phase, %).**
> > >
> > > 	Dataset                 CIFAR-100
> > >     phase          1      2       3       4       5
> > >     __________________________________________________
> > >     iCaRL-CNN    84.1   70.48   58.37   49.26   41.84
> > >     + IIB        84.1   77.1    68.35   59.06   53.03
> > >     __________________________________________________
> > >     iCaRL-NME    83.35  71.6    63.42   58.07   53.71
> > >     + IIB        83.35  77.75   69.42   60.85   56.50
> > >     __________________________________________________
> > >     SSIL-CNN     84.1   46.47   42.32   44.15   37.19
> > >     + IIB        84.1   72.87   64.92   57.29   52.14
> > >     __________________________________________________
> > >     SSIL-NME     83.35  71.5    64.4    57.89   53.16
> > >     + IIB        83.35  77.77   67.92   59.97   54.7
> > >     __________________________________________________
> > >
> > > Thank you for the suggestion, the IIB method really improves the performances significantly. As shown in Tab. 7,  iCaRL + IIB and SS-IL +IIB outperform iCaRL and SS-IL by large margins, about [1.6%, 15%] on the output of CNN or the nearest-mean-exemplar (NME) after completing the last learning phase.

---

> > > > ### Comment · Reviewer_JLqd · 2022-08-08
> > > > **A concern still remains**
> > > >
> > > > Thanks for the responses.
> > > > However, I still believe Table 7 is not a complete experiment to demonstrate the replaceability of the proposed IIB. As far as I know, both iCaRL and SSIL do not utilize CBF (class-balanced fine-tuning) process. For completeness, the authors also need to provide the results of **iCaRL + CBF and SSIL + CBF**. Or, another option is to provide the result of **(EEIL without CBF) + IIB** vs **(EEIL with CBF)**, since the authors used EEIL as the baseline in Table 3 in the main paper.

---

> > > > > ### Author Response · Authors · 2022-08-08
> > > > > **Additional experiments on the IIB method**
> > > > >
> > > > > Thanks for the comments. We conducted additional experiments based on the comments. EEIL[9]  fine-tunes the newly trained model with a balanced dataset (referred to as eeil-CBF), we replaced the eeil-CBF with IIB method at the second training stage. EEIL usually uses the output of CNN while iCaRL uses the output of NME. Here, we report both of the outputs of CNN and NME to compare the IIB method and eeil-CBF. The results are shown in Tab. 8. As shown in Tab. 8, both eeil-CBF and IIB improve the performances of baselines, iCaRL, SS-IL, and the baseline (EEIL without CBF). From Tab. 8, we can find that eeil-CBF brings about more improvements on the outputs of CNN while IIB improves the outputs of NME more significantly. Moreover, iCaRL-NME + IIB achieves the best performance. Overall, the results suggest that the IIB method is more helpful to learn good feature representation, compared with eeil-CBF.
> > > > >
> > > > > **Table 8. Results of experiments on replacing eeil-CBF with IIB method on CIFAR-100 with 5 phases in Base-0 protocol (Average Accuracy on each incremental phase, %), the buffer size is fixed to 2000, the baseline in the table is EEIL without CBF, * denotes using the results in EEIL[9].**
> > > > >
> > > > > 	Dataset                 CIFAR-100
> > > > >     phase         1      2     3      4      5
> > > > >     iCaRL*(NME)  83.6   71.2   62.6   56.5   53.9
> > > > >     EEIL*(CNN)   83.6   76.1   65.1   58.9   53.2
> > > > >     __________________________________________________
> > > > >     iCaRL-CNN    84.1   70.48  58.37  49.26  41.84
> > > > >     + eeil-CBF   84.1   72.92  65.33  59.75  54.14
> > > > >     + IIB        84.1   77.1   68.35  59.06  53.03
> > > > >     __________________________________________________
> > > > >     iCaRL-NME    83.35  71.6   63.42  58.07  53.71
> > > > >     + eeil-CBF   83.35  72.35  64.16  57.94  53.39
> > > > >     + IIB        83.35  77.75  69.42  60.85  56.50
> > > > >     __________________________________________________
> > > > >     SSIL-CNN     84.1   46.47  42.32  44.15  37.19
> > > > >     + eeil-CBF   84.1   73.28  65.77  59.54  53.85
> > > > >     + IIB        84.1   72.87  64.92  57.29  52.14
> > > > >     __________________________________________________
> > > > >     SSIL-NME     83.35  71.5   64.4   57.89  53.16
> > > > >     + eeil-CBF   83.35  72.45  64.45  58.03  53.58
> > > > >     + IIB        83.35  77.77  67.92  59.97  54.7
> > > > >     __________________________________________________
> > > > >     Baseline-CNN 84.1   71.6   59.01  52.6   42.84
> > > > >     + eeil-CBF   84.1   75.05  66.35  59.79  54.02
> > > > >     + IIB        84.1   72.35  65.88  59.70  53.69
> > > > >     __________________________________________________
> > > > >     Baseline-NME 83.35  73.42  64.43  58.14  52.68
> > > > >     + eeil-CBF   83.35  73.3   65.4   59.42  53.35
> > > > >     + IIB        83.35  77.95  67.82  59.85  53.99

---

> > > > > > ### Comment · Reviewer_JLqd · 2022-08-10
> > > > > > **Thanks for the feedback from the authors**
> > > > > >
> > > > > > Thanks for the feedback from the authors.
> > > > > > It seems that the proposed IIB training can replace CBF to some extent, while it does not show better performance than CBF for the CNN settings.
> > > > > > Even though both proposed methods (Re-MKD and IIB loss) may not be super-novel, one can consider IIB training procedure as a possible option to replace CBF.
> > > > > > I think it is a valuable finding, and therefore I will keep my positive rating for this submission.

---

### Official Review · Reviewer_1CA1 · 2022-07-13

**Rating:** 4
**Confidence:** 4
**Soundness:** 2 fair
**Presentation:** 2 fair
**Contribution:** 2 fair

**Summary:**

In this paper, the authors find that the main problem with the rehearsal methods is decision boundary overfitting to new classes. To solve this problem, they propose to combine the mixup and re-sampling strategy to synthesize adequate data used in knowledge distillation. Extensive experiment results are provided.

**Questions:**

Please address the issues raised in the "weaknesses" part.

**Limitations:**

It would be better to add a section to discuss the potential negative societal impact.

**Strengths And Weaknesses:**

### Strengths

- The proposed method is technically sound.

- This paper is well-organized and easy to follow.

- Extensive experimental results and visualizations are provided.

&nbsp;

### Weaknesses

- **The authors only provide results on small-scale datasets, e.g., CIFAR-100.** However, it is quite common to evaluate class-incremental learning on large-scale datasets. For example, iCaRl [12], BiC [8], LUCIR [10], Mnemonics [51], and PODNet-CNN [52] all provide result on ImageNet (full size, 1000 classes). Thus, it is not reasonable to ignore large-scale datasets for a top-tier conference submission.

&nbsp;

- **The technical novelty of this paper is somewhat limited.** This paper is based on re-sampling and mixup. Both strategies have already been widely used in many related topics, such as long-tailed recognition and few-shot learning. Thus, the technical contributions of this paper are somewhat limited.

&nbsp;

- **The state-of-the-art method is compared in Table 1.** For example, [A] archives much better performance than other baselines, but it is not compared in this paper. It seems the proposed method cannot achieve better performance than [A].

&nbsp;

[A] Yan, Shipeng, Jiangwei Xie, and Xuming He. "Der: Dynamically expandable representation for class incremental learning." Proceedings of the IEEE/CVF Conference on Computer Vision and Pattern Recognition. 2021.


&nbsp;

=== **After rebuttal** ===

Thanks for the active responses from the authors. The authors provide some new results in the rebuttal period and address some of my concerns. However, my major concerns about large-scale datasets and the technical novelty still remain. Thus, I tend to keep my initial rating.

---

> ### Author Response · Authors · 2022-08-02
> **Reply point-by-point to the comments**
>
> **Q1 (Only provide results on small-scale datasets)**
>
>  We appreciate the reviewer's comment.  Although we agree with this comment, please note that there is much room for improvement on small-scale datasets, and many latest works also conduct experiments on small image datasets in CIL, e.g., DualAug( NeurIPS 2021)}, Co2L ( ICCV 2021), etc. We conduct experiments to compare our method with DualAug and Co2L. The results are shown in Tabs.1 and 2 . As shown in Tabs.1 and 2, EDBL outperforms them by large margins, about [1.5\%, 8\%] in class incremental learning (Class-IL, CIL) setting on CIFAR-10, CIFAR-100, and Tiny-Imagenet. EDBL also performs well in the task incremental learning (Task-IL) setting, we can find that EDBL surpasses Co2L, DER, and DER++ (Dark experience for general
> continual learning: a strong, simple baseline, NeurIPS2020) by about [3\%, 14\%] in the Task-IL setting on CIFAR-10 and Tiny-Imagenet.
>
> We also want to point out that due to the GPU memory required by re-sampling Mixup during training, it is hard to accomplish Re-MKD on ImageNet-1000 with the shape of (224,224) in the short rebuttal period time. We estimate the training time of the experiment on ImageNet-1000 to be more than 15 days on 4 NVIDIA 3090. We plan to conduct experiments on ImageNet-1000 with Re-MKD and
> EDBL and report the results in the final version of the paper.
>
> **Table 1 Results of experiments conducted according to the protocol in Co2L[1]. Buffer size is 500, the incremental learning phases are 5 for CIFAR-10 and 10 for Tiny-Imagenet, * denotes using the results in [1]. (Average Acc. after completing the last phase, \%)**
>
> 	Dataset    CIFAR-10  CIFAR-10  Tiny-Imagenet  Tiny-Imagenet
>     Scenarios  Class-IL  Task-IL 	Class-IL	Task-IL
>     A-GEM*      22.67     89.48      8.06           25.33
>     iCaRL*      47.55     88.22      9.38           31.55
>     FDR*        28.71     93.29      10.54          49.88
>     DER*        70.51     93.40      17.75          51.78
>     DER++*      72.70     93.88      19.38          51.91
>     Co2L*       74.26     95.90      20.12          53.04
>     EDBL-NME    77.01     96.86      28.06          67.20
>
> **Table 2 Average incremental accuracy (\%) on Base-half experiments. Models with an asterisk * denotes using the results in DualAug.**
>
> 	Base-half    	CIFAR-100 CIFAR-100  Tiny-Imagenet    Tiny-Imagenet
>     phases              5	     10 	   5	         10
>     iCaRL*            59.67    56.13         48.98        39.27
>     BiC*              61.14    58.4          49.23        47.67
>     LUCIR*            63.17	   60.14	 49.31	      47.56
>     SSIL-NME          64.94    60.99	 48.93	      45.74
>     DualAug*          65.3     57.85         47.1         44.75
>     EDBL-CNN(ours)    66.57    62.06         52.43	      53.80
>     EDBL-NME(ours)    66.65	   64.73	 50.99	      49.97
>
> **Q2 (Re-sampling and Mixup}**
>
> Re-sampling and Mixup have already been widely used in many related topics, however, please note they have been mainly applied in classification while we apply them to tackle long-tail knowledge distillation (KD training with class imbalanced data, referred to as LT-KD) in CIL. Moreover, Beyer et al.[16] and Wang et al. [44] apply mixup to improve KD, but they use the data that are independent-identically-distributed (iid) with the original training data to make KD training, they can't verify the effectiveness of their methods in the CIL setting, where the training data are long-tailed and OOD.
> In this work, our contribution lies in validating the effectiveness of re-sampling and Mixup between old classes and new classes to tackle KD in the CIL setting. In addition, because KD with the long-tailed and OOD data is not the typical LT-KD training, so we first apply re-sampling and mixup to improve the performance of KD with OOD data in the first training stage. Then in the second training stage, which has become a typical LT-KD training, we develop the IIB factor especially the KD weighting factor to re-weight high-influenced samples in the balanced training to tackle long tail KD to fine-tune the new model. To the best of our knowledge, this is the first time that a work  focuses on KD with OOD data and the LT-KD problem in the CIL setting.
>
> **Q3 (Compare with Der[A])**
>
> Der employs a technique to dynamically expand the network and then compress it by pruning techniques. Der focuses on reliving catastrophic forgetting and its scalability is wholly based on the pruning technique. However, our work focuses on tackling KD with OOD data and LT-KD to alleviate forgetting. We consider this comparison interesting, and we plan to conduct experiments and discuss the influence of EDBL on knowledge transferring or model compressing by using the long-tail and OOD data for future work.
>
> **Q4 (Potential negative societal impact)**
>
> Thank you for the suggestion,  we will add a new section to discuss the potential negative societal impact in the final version of this paper.

---

> > ### Comment · Reviewer_1CA1 · 2022-08-02
> > **Thanks for the feedback from the authors.**
> >
> > Thanks for the feedback from the authors.
> >
> > However, I tend to keep my initial rating, "reject". The reasons are as follows,
> >
> > - My suggestion is to provide the results on ImageNet-Full (1000 classes). Because my existing paper, e.g., iCaRl [12], BiC [8], LUCIR [10], Mnemonics [51], and PODNet-CNN [52] all provide result on ImageNet (full size, 1000 classes). However, the authors provide the results on two other datasets, CIFAR-10 and Tiny-ImageNet. The first one is still a small-scale dataset, and the second one has a very small image size. The most important reason is that the provided results are still not comparable with the existing papers, e.g., [8, 10, 12, 51, 52].
> >
> > &nbsp;
> >
> > - I don't think "validating the effectiveness of re-sampling and Mixup between old classes and new classes to tackle KD in the CIL setting" is a significant contribution to a top-tier conference like NeurIPS.
> >
> > &nbsp;
> >
> > - For my Q3, the authors' response is that they can do it as a part of future work. I don't think it is an informative response. Maybe I can consider accepting their future work if I have the opportunity to review it.

---

> > > ### Author Response · Authors · 2022-08-04
> > > **Thanks for Reviewer 1CA1's feedback**
> > >
> > > We appreciate your suggestions. Here are our comments on your concerns.
> > >
> > > **Question: Experiments on ImageNet-Full (1000 classes)**
> > >
> > > Answer: Thank you very much for this question. On the one hand, our method has not only achieved SOTA performances on CIFAR 10, CIFAR 100, and Tiny-ImageNet in the CIL setting, but also outperforms all the baselines by large margins in the Task-IL setting on CIFAR 10 and Tiny-ImageNet. On the other hand, the Tiny-ImageNet classification challenge is similar to the classification challenge in the full ImageNet ILSVRC. Tiny-ImageNet contains 200 classes for training. Each class has 500 images. The test set contains 10,000 images. All images are 64x64 colored ones, where the list of the used ids comes from the original full set of ImageNet (Please refer to the URL: https://www.kaggle.com/competitions/tiny-imagenet/overview). We believe that a similar result would be got in ImageNet-Full and we can provide the performances in our final version of this paper. The main reason is that we estimate the training time of the experiment on ImageNet-1000 to be more than 15 days on 4 NVIDIA 3090 while only limited time for us to run new experiments on ImageNet-1000 in one week (rebuttal period).
> > >
> > >
> > > **Question: Method Novelty**
> > >
> > > Answer: As for the novelty of Re-MKD, Re-MKD is one of our contributions and it is mainly used in the first training stage. We split the training into two stages. The first stage is used to transform the OOD-KD training (Knowledge distillation with OOD data, referred to OOD-KD) into a typical LT-KD training (Knowledge distillation with long-tailed data, referred to LT-KD). RE-MKD is mainly used in the first training stage to improve the performance of OOD-KD. Then in the second training stage, we attempt to tackle LT-KD to transfer knowledge from the head (new) classes to the tail (old) classes by the IIB method. To the best of our knowledge, this is the first time work focuses on KD with OOD data and the LT-KD problem in the continual learning setting. We argue that the training schedule and the IIB method also contribute to the continuous learning community.
> > >
> > >
> > > **Question: Potential negative societal impact**
> > >
> > > Answer: We are sorry for misunderstanding you. Actually, in Sec 6 of our paper, we have provided the limitations and broader impact. This paper focuses on class incremental learning for image recognition. If the CIL methodology is used in sensitive applications such as face recognition (private area), it may cause some potential negative societal impact.

---

> > > > ### Comment · Reviewer_1CA1 · 2022-08-08
> > > > **Thanks for the feedback from the authors.**
> > > >
> > > > Thanks for the active feedback from the authors. I am truly grateful for it.
> > > >
> > > > I ask for the results on ImageNet-1k because I hope all papers for CIL can share the same benchmarks and datasets. As I have mentioned in my review, lots of work on CIL, e.g., iCaRl [12], BiC [8], LUCIR [10], Mnemonics [51], and PODNet-CNN [52], has provided results on ImageNet-1k. If you think it is interesting to include the results on some new datasets, it is fine. However, you cannot ignore some popular datasets that have been widely applied. Otherwise, it would be very difficult for the following researchers to compare their work with yours.
> > > >
> > > > Besides, I am not asking for last-minute experiments. I think it would be better to submit the updated version to the next venue, e.g., ICLR.
> > > >
> > > > Based on the overall quality, I cannot strongly support this paper. Thus, my final rating is "borderline reject".

---

> > > > > ### Author Response · Authors · 2022-08-08
> > > > > **Thanks for Reviewer 1CA1's feedback**
> > > > >
> > > > > We agree with the proposal to use the same benchmarks and the same evaluation protocol for all continuous learning works, this is beneficial to the continuous learning community. One of the reasons why we use Tiny-ImageNet, CIFAR-10/100 instead of ImageNet-1k in this paper is that Tiny-ImageNet, CIFAR-10/100 are also widely used benchmarks in CL. Many previous works apply these data sets but not ImageNet-1k, e.g., Co2L(ICCV2021), [43], (Bring Evanescent Representations to Life in Lifelong Class Incremental Learning, CVPR2022),  etc.. We will open the source code, this is very convenient to validate our method on ImageNet-1k. Anyhow, thanks for the comments.

---

> > > > > > ### Comment · Reviewer_1CA1 · 2022-08-09
> > > > > > **Thanks for the feedback from the authors.**
> > > > > >
> > > > > > Thanks for the feedback from the authors.
> > > > > >
> > > > > > I decide to keep my initial rating. The major reasons are my concerns about benchmarks and technical novelty.

---

### Author Response · Authors · 2022-08-02
**Results of extra experiments  and references for rebuttal**

**We appreciate all reviewers for their comments.**  Here, we give the references and all the results of extra experiments during the rebuttal period.

**Table 1 Results of experiments conducted according to the protocol in Co2L[1]. Buffer size is 500, the incremental learning phases are 5 for CIFAR-10 and 10 for Tiny-Imagenet, * denotes using the results in [1]. (Average Acc. after completing the last phase, \%)**

	Dataset       CIFAR-10  CIFAR-10  Tiny-Imagenet  Tiny-Imagenet
    Scenarios     Class-IL  Task-IL   Class-IL	 Task-IL
    A-GEM*[2]      22.67     89.48      8.06           25.33
    iCaRL*[3]      47.55     88.22      9.38           31.55
    FDR* [4]       28.71     93.29      10.54          49.88
    DER*[5]        70.51     93.40      17.75          51.78
    DER++*[5]      72.70     93.88      19.38          51.91
    Co2L*[1]       74.26     95.90      20.12          53.04
    EDBL-NME(ours) 77.01     96.86      28.06          67.20

**Table 2 Average incremental accuracy (\%) on Base-half experiments. Models with an asterisk * denote using the results in DualAug.**

	Base-half    	CIFAR-100 CIFAR-100  Tiny-Imagenet    Tiny-Imagenet
    phases              5	     10 	   5	         10
    iCaRL[3]          59.67    56.13         48.98        39.27
    BiC[6]            61.14    58.4          49.23        47.67
    LUCIR[7]          63.17    60.14	 49.31	      47.56
    SSIL-NME[8]       64.94    60.99	 48.93	      45.74
    DualAug*[9]       65.3     57.85         47.1         44.75
    EDBL-CNN(ours)    66.57    62.06         52.43	      53.80
    EDBL-NME(ours)    66.65    64.73	 50.99	      49.97

[1] Hyuntak Cha, et al.. Co2l: Contrastive continual learning. ICCV, 2021

[2] Arslan Chaudhry, et al.. Efficient lifelong learning with a-gem. ICLR, 2018.

[3] Sylvestre-Alvise Rebuffi, et al.. icarl: Incremental classifier and representation learning. CVPR, 2017.

[4] Ari S. Benjamin, et al.. Measuring and regularizing networks in function space. ICLR, 2019.

[5] Pietro Buzzega, et al.. Dark experience for general continual learning: a strong, simple baseline. NeurIPS, 2020.

[6] Yue Wu, et al.. Large-scale incremental learning. CVPR, 2019.

[7] Saihui Hou, et al.. Learning a unified classifier incrementally via rebalancing. CVPR, 2019.

[8] Hongjoon Ahn, et al.. Ss-il: Separated softmax for incremental learning. ICCV, 2021.

[9] Fei Zhu, et al.. Class-incremental learning via dual augmentation. NeurIPS, 2021

**Table 3. Results of experiments on excluding the mixed data of the newly added classes in KD training (%).**

	Dataset                                CIFAR-100
    phase                     1     2       3     4       5
    Baseline-NME             84.8  73.3   65.67  59.09  54.47
    Baseline-NME +exclude    84.8  69.87  62.90  56.51  52.19
    EDBL-NME                 85.01 75.25  68.05  62.95  57.52
    EDBL-NME +exclude        84.8  63.95  58.38  52.62  48.55

**Table 4. Results of Re-MKD + CBF on CIFAR-100 with 5 phases in Base-0 protocol~(Average Accuracy on each incremental phase, %).**

	Dataset                 CIFAR-100
    phase                       1     2       3     4      5
    BiC                       84.8  74.02  66.7   61.5   56.5
    BiC+Re-MKD                84.8  71.73  59.36  57.59  53.51
    EEIL                      83.5  76.5   64.2   59.1   52.8
    EEIL+Re-MKD               84.8  71.85  64.78  58.14  52.84
    IIB-KD(One-stage)         83.5  69.47  60.3   53.15  48.7
    IIB-KD(One-stage)+Re-MKD  84.8  76.7   70.93  65.73  60.51

**Table 5. Results (Average Accuracy, %) of study on re-sampling rate.**

    Dataset                  CIFAR-10                           CIFAR-100
    phase          1     2      3     4       5        1      2      3      4      5
    Baseline-NME 99.3  85.15  64.27 58.39	61.24     84.8  69.52  61.42  55.52  50.02
          4      	               /                  84.8	69.77  63.4   58.2   53.91
          2      99.5  79.72  71.68	69.48	66.9      84.8	68.27  63.32  58.75  54.47
          1      99.5  86.87  75.33	74.14	71.27     84.8	72.17  65.05  58.99  55.27
         0.5     99.5  87.32  76.48	70.54	68.64     84.8	72.25  64.95  58.34  54.14
        0.25   	               /                  84.8	72.9   64.1   57.55  53.4

**Table 6. Average accuracy and Average forgetting rate (\%) after completing the last learning phase.**

    Base-0           CIFAR-10  CIFAR-10    CIFAR-100  CIFAR-100
      /                ACC       FGT 	   ACC	      FGT
    iCaRL             66.06     20.98         54.2       11.15
    BiC               65.97     19.58         56.5       10.77
    SSIL-NME          66.17      8.55	  53.54	     8.28
    EDBL-CNN(ours)    66.18      6.45	  60.51	     5.14
    EDBL-NME(ours)    68.77	    12.90	  57.52	     12.92

---

### Meta-Review · Area_Chair_PRjw · 2022-08-22

**Recommendation:** Reject
**Confidence:** Certain

**Metareview:**

The paper aims to use the mixup and re-sampling strategy to improve knowledge distillation. Reviewers find that the classification weighting factor is not a new task, and the technique and novelty is very limited. Because both the mixup and re-sampling strategy have been used in many settings. There is no techniqual contribution. The author is encouraged to improve the paper by considering the comments from the reviewers.

**Award:**

No

---

### Decision · Program_Chairs · 2022-09-14

Reject